# Spatiotemporally-resolved mapping of RNA binding proteins via functional proximity labeling reveals a mitochondrial mRNA anchor promoting stress recovery

Wei Qin[1,2], Samuel A. Myers[3,4], Dominique K. Carey[3], Steven A. Carr[3] & Alice Y. Ting [1,2✉]

Proximity labeling (PL) with genetically-targeted promiscuous enzymes has emerged as a powerful tool for unbiased proteome discovery. By combining the spatiotemporal specificity of PL with methods for functional protein enrichment, we show that it is possible to map specific protein subclasses within distinct compartments of living cells. In particular, we develop a method to enrich subcompartment-specific RNA binding proteins (RBPs) by combining peroxidase-catalyzed PL with organic-aqueous phase separation of crosslinked protein-RNA complexes ("APEX-PS"). We use APEX-PS to generate datasets of nuclear, nucleolar, and outer mitochondrial membrane (OMM) RBPs, which can be mined for novel functions. For example, we find that the OMM RBP SYNJ2BP retains specific nuclear-encoded mitochondrial mRNAs at the OMM during translation stress, facilitating their local translation and import of protein products into the mitochondrion during stress recovery. Functional PL in general, and APEX-PS in particular, represent versatile approaches for the discovery of proteins with novel function in specific subcellular compartments.

[1] Departments of Biology, Genetics, and Chemistry, Stanford University, Stanford, CA, USA. [2] Chan Zuckerberg Biohub, San Francisco, CA, USA. [3] The Broad Institute of MIT and Harvard, Cambridge, MA, USA. [4] La Jolla Institute for Immunology, La Jolla, CA, USA. ✉email: ayting@stanford.edu

Proximity labeling (PL) provides an alternative to traditional methods such as immunoprecipitation and biochemical fractionation for the proteomic analysis of protein inter-action networks and organelle components[1,2]. In PL, an engi-neered promiscuous labeling enzyme, such as APEX[3], TurboID[4], or BioID[5], is targeted via genetic fusion to a specific subcellular compartment or macromolecular complex. Addition of a small molecule substrate to live cells results in catalytic generation of a reactive biotin species (such as biotin-phenoxyl radical for APEX or biotin-AMP for TurboID) from the active site of the pro-miscuous enzyme, which diffuses outward to covalently tag proximal endogenous proteins. PL methods have been used to map organelle proteomes (mitochondrion, ER, lipid droplets, stress granules)[6], dynamic interactomes[7,8], and in vivo secretomes[9,10]. Recently, PL was extended to spatial mapping of transcriptomes in living cells[11–13].

While PL is a powerful technology for unbiased discovery of proteins or RNA in specific subcellular locales, in many cases one wishes to focus the search on a specific protein or RNA functional subclass. PL has not previously been used in the manner of activity-based protein profiling[14,15], for instance, to enrich specific enzyme families. If one could combine the spatiotemporal speci-ficity and live-cell compatibility of PL with targeted approaches that enrich proteins or RNA based on functional activity or covalent modifications ("functional proximity labeling"), it would enable the simultaneous assignment of localization, timing, and function to specific protein populations.

Here we demonstrate that the simple addition of a PL step is able to endow several functional enrichment strategies with nanometer-scale spatial resolution for subcellular organelles and compartments of interest. We demonstrate this for enrichment of subcellular phosphoproteins, O-linked-N-acetylglucosaminylated (O-GlcNAcy-lated) proteins, and RNA-binding proteins (RBPs). In particular, RBPs constitute an important and large (estimated ~10%[16]) func-tional subclass of the human proteome[17–19]. While methods for the global discovery of RBPs, such as oligo(dT) bead capture[20–23], metabolic RNA labeling[24,25], and organic-aqueous phase separation[26–28], are highly valuable, a richer understanding of RBP function could be achieved with spatially and temporally resolved approaches that report not only the proteins that have RNA-binding function but also where they are located within the cell and under what conditions. By combining APEX-mediated proximity biotiny-lation with organic-aqueous phase separation to enrich crosslinked protein–RNA complexes ("APEX-PS"), we are able to map RBPs in the nucleus, as well as RBPs in non-membrane enclosed regions that are difficult or impossible to purify by biochemical fractionation—the nucleolus and the outer mitochondrial membrane (OMM).

We also used APEX-PS to study RBPs in relation to mito-chondria and stress. We discovered that an OMM-localized RBP, SYNJ2BP (Synaptojanin-2-binding protein), helps to safeguard important nuclear-encoded mitochondrial mRNAs under a variety of stress conditions. This retention mechanism facilitates the local translation of these transcripts during stress recovery, leading to rapid restoration of OXPHOS activity and overall mitochondrial function. Functional PL therefore provides a powerful approach to dissect functional events with spatio-temporal precision and drive the discovery of novel biology.

## Results

### Compartment-specific enrichment of phosphorylated and O-GlcNAcylated proteins.
In a given organelle or subcellular compartment, many different protein types function in concert to carry out the different sensing, signaling, regulatory, and meta-bolic needs of that compartment[29]. We wondered whether it would be possible to combine the PL workflow with various methods for functional protein enrichment in order to extract additional information from proteomics experiments (Fig. 1a). We selected the fastest PL enzyme available: APEX2, an engi-neered peroxidase that catalyzes the one-electron oxidation of biotin-phenol conjugates[30]. The radical product of this oxidation forms covalent adducts with electron rich protein sidechains such as tyrosine and has a half-life of <1 millisecond, resulting in labeling radii <10 nm[3]. Labeling is initiated by the addition of biotin-phenol and $H_2O_2$ to live cells and typically performed within 20 seconds to 1 min[3].

To begin, we explored the combination of APEX-based PL with either phosphoprotein enrichment via immobilized metal affinity chromatography (IMAC)[31], or O-GlcNAcylated protein enrichment via wheat germ agglutinin (WGA) affinity chromatography[32] (Fig. 1b and Supplementary Fig. 1a). IMAC enrichment is widely used for phosphoproteomics[31,33], and phosphorylation represents an impor-tant post-translational modification (PTM) involved in the regula-tion of many dynamic cellular processes[34,35]. O-GlcNAcylation on serine and threonine sidechains of intracellular proteins is another type of ubiquitous PTM[36], which can be recognized and captured by WGA[32,37]. Considering that phosphorylation and O-GlcNAcylation are widely distributed in various compartments including the nucleus and stress granules[38,39], the ability to selectively probe these region-specific PTMs should facilitate our understanding towards their functional roles.

To evaluate functional PL on nuclear phosphorylation and O-GlcNAcylation, we prepared HEK293T cells stably expressing nucleus-targeted APEX2-NLS, and performed live-cell labeling with biotin-phenol for 1 min. Separately, we verified that this 1 min labeling in the presence of $H_2O_2$ did not alter global phosphoryla-tion or O-GlcNAcylation levels (Supplementary Fig. 1b, c). We used IMAC to first capture total phosphorylated proteins and the eluates were subjected to streptavidin-based enrichment of biotinylated phosphoproteins (Supplementary Fig. 1d). Impor-tantly, nuclear phosphoproteins were not detected when the cell lysates were treated with phosphatase, or $H_2O_2$ was omitted (Supplementary Fig. 1e). To probe the specificity of this dual enrichment protocol, we performed western blotting for known nuclear phosphoproteins as well as phosphoproteins in other subcellular regions (Fig. 1c). As expected, only the nuclear phosphoproteins SMAD2 (Mothers against decapentaplegic homo-log 2) and Histone H3 were enriched, while the mitochondrial phosphoprotein TOMM20 (Mitochondrial import receptor sub-unit TOM20 homolog) and ER phosphoprotein CANX (Calnexin) were not detected. Similarly, nuclear O-GlcNAcylated proteins were specifically captured by WGA and streptavidin tandem pulldown. We detected the nuclear O-GlcNAcylated proteins SP1 (Transcription factor Sp1) and FBL (Fibrillarin), but not the mitochondrial O-GlcNAcylated protein SDHA (Succinate dehy-drogenase complex flavoprotein subunit A) or the nuclear non-glycosylated protein NHP2L1 (NHP2-like protein 1) (Fig. 1d).

Given the 1 min temporal resolution of APEX labeling, we wondered whether dynamic phosphorylation events could be monitored by functional PL. Upon TPA (12-O-Tetradecanoylphor-bol-13-acetate) stimulation, cytosolic ERK2 (Extracellular signal-regulated kinase 2) is rapidly phosphorylated to induce its nuclear translocation[40] (Fig. 1e). Functional PL revealed that the phosphor-ylation level of ERK2 dramatically increases in the nucleus but not in the cytosol after 15 min of TPA treatment (Fig. 1f). This demonstration shows that functional PL can be used to investigate dynamic PTMs in a spatiotemporally resolved manner.

### Development of APEX-PS for functional proximity labeling of subcellular RBPs.
With the successful enrichment of subcellular PTMs, we turned our attention to RBPs, a large functional subclass

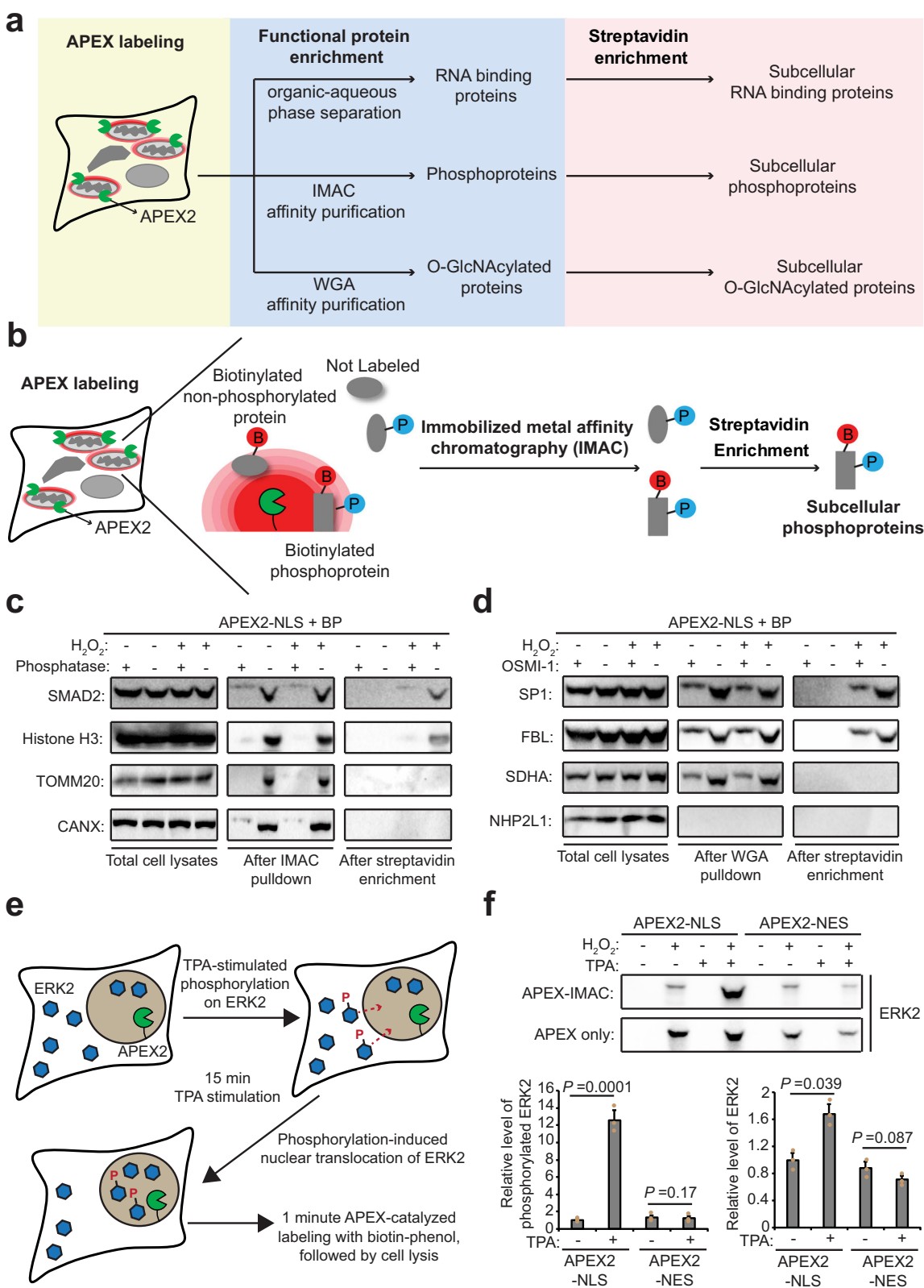

of the human proteome. Noncovalent RNA–protein interactions are pervasive in both transient and stable macromolecular complexes underlying transcription, translation, and stress response[16,41]. A spatiotemporally resolved approach for RBP discovery could yield valuable insights into organelle- or region-specific questions concerning the dynamics of RNA–protein interactions.

To adapt APEX-based PL for spatiotemporally resolved enrichment of RBPs, we envisioned coupling the 1 min live-cell APEX biotinylation with RNA–protein crosslinking using formaldehyde (FA) or UV (Fig. 2a), and then enriching crosslinked RNA–protein complexes via orthogonal organic phase separation (OOPS), an unbiased RBP profiling approach with superb sensitivity[26,42]. In OOPS, proteins partition to the

**Fig. 1 Development and validation of functional proximity labeling to study subcellular phosphorylation and O-GlcNAcylation. a** The workflow of functional PL that combines APEX-catalyzed PL with functional protein enrichment (e.g., phase separation, IMAC, or WGA) and streptavidin bead capture to enrich subcellular protein subclasses. **b** Procedure combining APEX-catalyzed PL with IMAC to enrich subcellular phosphoproteins. Red B, biotin. Blue P, phosphorylation. **c** Western blot detection of known nuclear phosphoproteins in lysates from HEK cells expressing APEX2-NLS. After 1 min of biotin-phenol labeling, cells were lysed and subjected to IMAC and streptavidin enrichment as shown in **b**. SMAD2 and Histone H3 are true-positive nuclear phosphoproteins. TOMM20 and CANX are true negative mitochondrial and ER phosphoproteins, respectively. Phosphatase treatment of cell lysate removes phosphorylation. **d** Specificity validation for nuclear O-GlcNAcylated proteins captured by WGA and streptavidin tandem enrichment. Lysates generated from HEK cells expressing APEX2-NLS, labeled for 1 min with biotin-phenol. SP1 and FBL are true-positive nuclear O-GlcNAcylated proteins. SDHA is a true negative mitochondrial O-GlcNAcylated protein, and NHP2L1 is a true negative nuclear protein that is not O-GlcNAcylated. OSMI-1 is an O-GlcNAc transferase inhibitor that reduces global O-GlcNAcylation level. **e** Model of 12-O-tetradecanoylphorbol-13-acetate (TPA)-stimulated phosphorylation and nuclear translocation of ERK2. **f** Western blot analysis of phosphorylated ERK2 (APEX-IMAC) and total ERK2 (APEX only) levels in the nucleus (APEX2-NLS) and cytosol (APEX2-NES). Statistical analysis (Two-sided Student's $t$-test) shown below. Values represent means ± SD from three biological replicates.

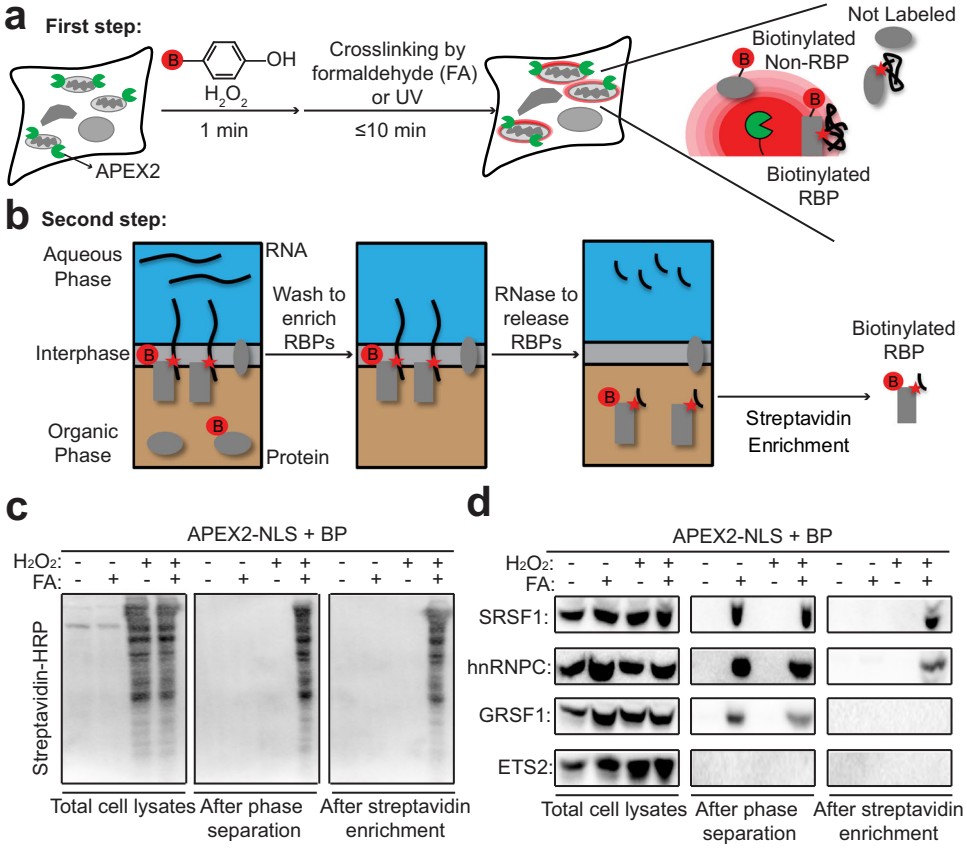

**Fig. 2 Development and validation of APEX-Phase Separation (APEX-PS) for enrichment of subcellular RNA-binding proteins (RBPs). a** In the first step of APEX-PS, 1 min APEX-catalyzed biotinylation is carried out, followed by 10 min of RNA–protein crosslinking in living cells. Red B, biotin. RBP, RNA-binding protein. **b** In the second step of APEX-PS, phase separation of cell lysate from **a** localizes crosslinked protein–RNA complexes to the interphase, while free proteins and RNA localize to the organic and aqueous phases, respectively. After washing, RNA-crosslinked proteins are released from the interphase by RNase treatment and subjected to streptavidin bead-based enrichment of biotinylated proteins. **c** Streptavidin blotting reveals enrichment of nuclear RBPs by APEX-PS. Biotinylation and formaldehyde crosslinking were performed in HEK293T cells expressing nuclear-localized APEX (APEX-NLS). Streptavidin blotting was performed on total cell lysate (left), samples after phase separation (middle), and samples after both phase separation and streptavidin enrichment (right). **d** Western blot detection of known nuclear RBPs in samples from **c**. SRSF1 and hnRNPC are true-positive nuclear RBPs, GRSF1 is a mitochondrial RBP, and ETS2 is a nuclear protein that does not bind RNA (non-RBP).

organic layer and RNAs partition to the aqueous layer, while crosslinked RNA–protein complexes partition to the interphase (Fig. 2b). After washing, RBPs can then be specifically released to the organic layer by RNase digestion, and extracted from this layer. OOPS has been used to identify mammalian RBPs on a large scale, albeit with loss of spatial information[26].

To test and optimize tandem APEX and OOPS ("APEX-PS"), we first focused on nuclear RBPs, which constitute a well-studied class of subcellular RBPs that have been characterized by

fractionation-based methods, such as serIC (Serial RNA interactome capture)[43] and RBR-ID (RNA-binding region identification)[44]. APEX-NLS labeling was performed with biotin-phenol for 1 min and then the cells were treated with FA for 10 min. The cell lysate was subjected to OOPS followed by streptavidin bead enrichment. Figure 2c shows that many biotinylated proteins were recovered by this procedure but not in negative controls omitting $H_2O_2$ (which suppresses APEX-catalyzed biotinylation) or FA (which prevents RNA–protein

crosslinking and hence recruitment of proteins to the interphase). To probe the identities of nuclear APEX-PS enriched proteins, we performed western blotting of enriched lysates (Supplementary Fig. 2a) for known nuclear RBPs as well as off-target protein markers (Fig. 2d). We detected the nuclear RBPs hnRNPC (heterogeneous nuclear ribonucleoproteins C1/C2) and SRSF1 (serine/arginine-rich splicing factor 1), while the mitochondrial RBP GRSF1 (G-rich sequence factor 1) and nuclear non-RBP ETS2 (C-ets-2) were not detected. Notably, GRSF1 was enriched after the PS step, because it is an RBP, but not after streptavidin bead enrichment, because it is not a nuclear protein and thus not biotinylated by APEX-NLS.

We also tested the APEX-PS workflow with UV instead of FA to crosslink RNA and proteins (Supplementary Fig. 2b). UV crosslinking, used in most RBP profiling studies, is thought to be more spatially specific than FA crosslinking, although FA has been successfully applied in multiple RBP-capture strategies including CHART[45], ChIRP[46,47], and CAPRI[23]. Moreover, FA crosslinking has been widely applied to map DNA–protein interactions[48–50] with high specificity. When we repeated APEX-PS using UV crosslinking, we again observed high compartment specificity and RBP specificity, as expected (Supplementary Fig. 2c). However, compared to FA, the capture efficiency with UV crosslinking was greatly reduced (Supplementary Fig. 2d, e) while the RBP specificity was unchanged; for example, both FA and UV captured RBPs in an RNase-dependent manner and both could discriminate between RBPs and abundant non-RBPs such as Histone H3 and ACTB (Supplementary Fig. 2e). Due to its high specificity combined with the much higher recovery efficiency, we selected FA as our RNA–protein crosslinking strategy for proteome-scale APEX-PS in this study. However, individual hits were validated by the more stringent UV-based APEX-PS.

To optimize APEX-PS, we also tested reversing the APEX and FA crosslinking steps. In a side-by-side comparison, we found that both protocols are highly sensitive, able to enrich true positives, but the APEX-first method is more specific, able to discriminate against the mitochondrial RBP GRSF1 when APEX is localized to the nucleus (Supplementary Fig. 2f). It is likely that when FA crosslinking is performed first, it compromises cellular membranes, enabling APEX-generated radicals to leak into other compartments and tag distal proteins. The APEX-then-FA protocol was also shown to be superior in our previous APEX-RIP study, which also employs FA crosslinking[51].

We utilized western blotting of specific protein markers to compare nuclear APEX-PS with nuclear fractionation[52] followed by OOPS[26]. Supplementary Figure 2g shows that both methods were highly specific but APEX-PS gives somewhat higher recovery efficiency for nuclear RBPs.

To test the generality of APEX-PS, we repeated it in four additional subcellular compartments: the cytosol, ER membrane facing cytosol, outer mitochondrial membrane facing cytosol, and nucleolus (Supplementary Fig. 3a). All of these gave $H_2O_2$- and FA-dependent signals in streptavidin blots of whole-cell lysates, with distinct banding patterns or "fingerprints" unique to their compartments (Supplementary Fig. 3b-e).

**Proteomic profiling of nuclear RBPs by APEX-PS**. Having optimized and validated APEX-PS in the nucleus by western blotting, we proceeded to a multiplexed, TMT-based proteomics experiment that included both nucleus-targeted APEX2-NLS and nucleolus-targeted APEX2-NIK3x (the latter discussed below). The experimental design (Fig. 3a) consisted of three biological replicates of each construct expressed in HEK293T cells, alongside three controls with APEX2, $H_2O_2$, or FA omitted. Imaging

showed correct localization of both APEX2-NLS (V5 staining) and APEX-biotinylated proteins (neutravidin staining) (Fig. 3b). After phase separation, streptavidin enrichment, and on-bead tryptic digestion to peptides, each sample was chemically tagged with a unique TMT label. The samples were then pooled and analyzed by liquid chromatography-tandem mass spectrometry analysis (LC-MS/MS).

A total of 1782 proteins were detected, each with two or more unique peptides (Fig. 3c and Supplementary Data 1). We observed excellent correlation across the three biological replicates (Supplementary Fig. 4a, b). To filter the data and arrive at our final nuclear RBPomes, we used two different approaches. In our "pairwise ROC approach"[53,54], we paired each replicate with a negative control to calculate TMT ratios. We then used a true-positive list of known nuclear RBPs and a false-positive list of mitochondrial proteins (which should not be biotinylated by APEX2-NLS) to generate receiver operating characteristic (ROC) curves for each replicate (Supplementary Fig. 4c). We applied the TMT ratio cutoff that maximized the difference between true-positive rate (TPR) and false-positive rate (FPR) for each dataset (Supplementary Fig. 4d). From the three resulting protein lists, we included in our "nuclear RBPome" the 863 proteins that were enriched in two or more replicates. Because phase separation is known to enrich glycosylated proteins due to the hydrophilicity of glycans[26], we manually removed glycoprotein contaminants from our dataset (the 72 removed glycoproteins can be seen in Tab 5 of Supplementary Data 2). Our final nuclear RBPome consists of 791 proteins and can be seen in Supplementary Data 2 (**"nuclear RBPome1"**).

The second approach we used to filter the MS data consists of two-sample $t$-tests to compare APEX-NLS replicates to background controls (i.e., omit APEX- and omit $H_2O_2$ negative controls) (Supplementary Fig. 5a). After removing glycoproteins from the significantly enriched proteins (adj. $p$-value < 0.05, fold change > 1), we obtained a final proteome size of 230 proteins (Supplementary Data 2; **"nuclear RBPome2"**); 224 of these proteins were also present in our nuclear RBPome1 (Supplementary Fig. 5b).

**Analysis of nuclear RBPomes generated by APEX-PS**. Gene Ontology (GO) analysis of both nuclear RBPomes showed enrichment of RNA-related terms including mRNA splicing and processing (Fig. 3d and Supplementary Fig. 5c). GO cellular component (GOCC) analysis showed significant enrichment of nucleus-related terms, as expected (Fig. 3e and Supplementary Fig. 5d). In order to further evaluate the specificity of the nuclear RBPome datasets, we compiled a list of 4925 human RBPs that have been previously annotated by RBP profiling methods[20,21,24,26–28,55–57] as well as 6889 nuclear proteins annotated by GOCC. Both our nuclear RBPome1 and nuclear RBPome2 show high RNA-binding (~92%) and nuclear (~74%) specificity (Supplementary Fig. 5e, f). However, our nuclear RBPome1 was much more sensitive than our nuclear RBPome2, because more true-positive nuclear RBPs were detected in the former (46%) than in the latter (20%) (Supplementary Fig. 5g). Our list of 155 true positives are high-confidence, literature-validated nuclear RBPs with functions in ribosome biogenesis, splicing, transcription, and other essential processes (Supplementary Data 2). Given the similar specificity but greater sensitivity of nuclear RBPome1 compared to nuclear RBPome2, we selected the former dataset for further analysis.

We compared our nuclear RBPome1 dataset to nuclear RBP datasets obtained by other methods (Fig. 3f-h). serIC[43] combines nuclear fractionation with repeated oligo(dT) RBP purification; 343 nuclear RBPs were enriched by this method from $10^9$ cells, 20

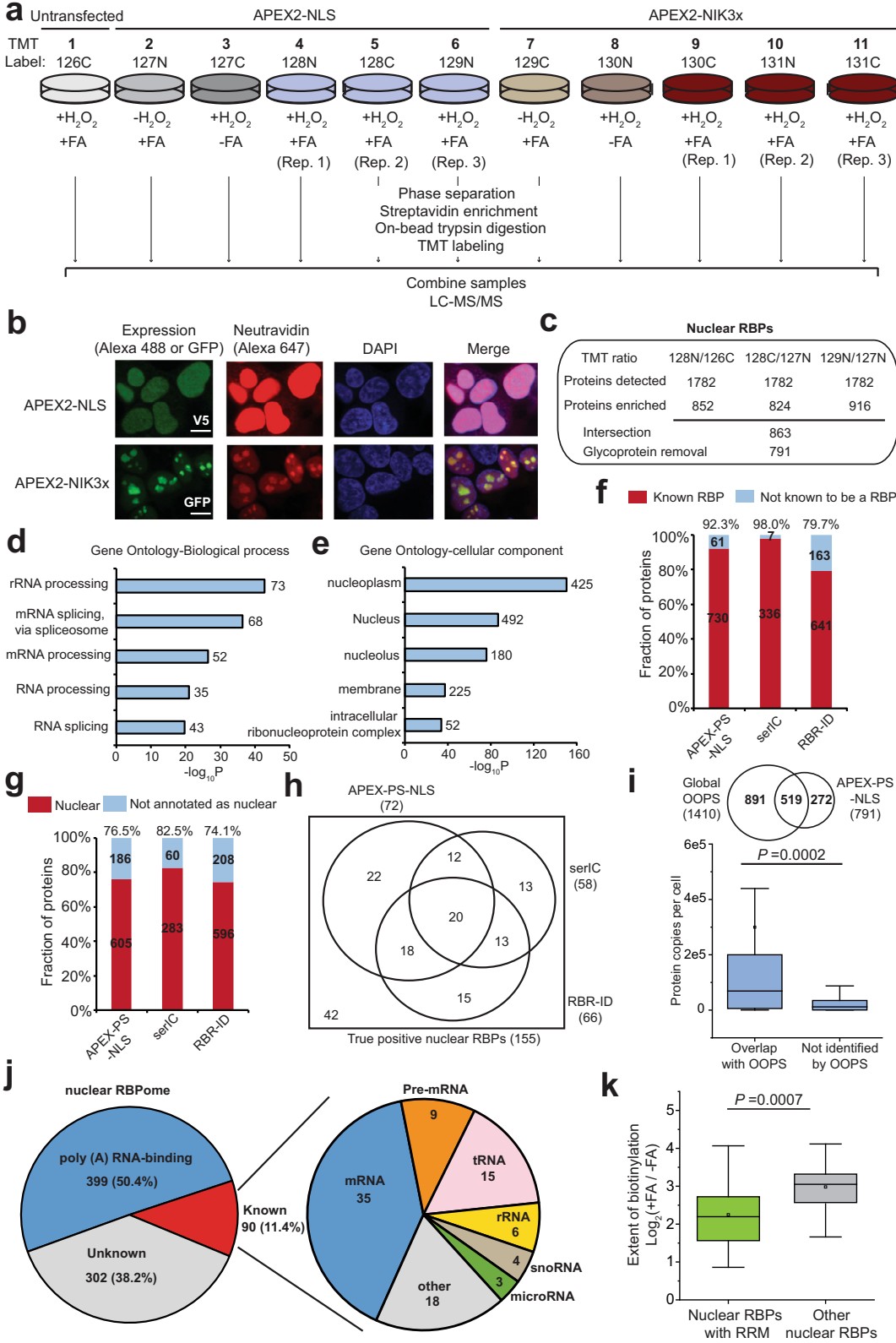

times more material than was used for nuclear APEX-PS. RBR-ID[44] is based on depletion of parent peptide signatures in MS due to RNA–protein crosslinking. Eight hundred four nuclear RBPs were identified from purified mouse embryonic stem cell nuclei by this approach. We found that serIC was extremely specific (98.0% RBP specificity and 82.5% nuclear specificity) but less sensitive (37.4%) than APEX-PS, missing RBPs that bind to non-

polyadenylated RNAs (e.g., tRNA ligases) for example (Supplementary Fig. 6a). RBR-ID was more sensitive than SerIC (42.6%) but much less specific (79.7% RBP specificity, 74.1% nuclear specificity). Our APEX-PS dataset, quantified in the same manner, showed high specificity (92.3% RBP specificity and 76.5% nuclear specificity) and the highest sensitivity of the three methods (46.5%). The true specificity of APEX-PS may be even

**Fig. 3 Proteomic profiling of nuclear RBPs by APEX-PS. a** Experimental design and labeling conditions for TMT-based proteomics. HEK293T cells stably expressing nuclear APEX2-NLS (samples 2–6) or nucleolar APEX2-NIK3x (samples 7–11) were subjected to proximity biotinylation and FA crosslinking. The control samples omitted enzyme (sample 1), $H_2O_2$ (sample 2 and 7), or FA (3 and 8). **b** Confocal fluorescence imaging of APEX2 localization (V5 or GFP) and biotinylation activity (neutravidin-Alexa 647) in the nucleus (top) and nucleolus (bottom). Live-cell biotinylation was performed for 1 min before fixation. DAPI stains nuclei. Scale bars, 10 μm. **c** Numbers of proteins remaining after each step of filtering the mass spectrometry data using the "pairwise ROC strategy". The final "nuclear RBPome1" obtained by APEX-PS has 791 proteins (Supplementary Data 2). **d** GO biological process analysis of nuclear RBPs identified by APEX-PS. The number of proteins in each GO term is shown. **e** GO cellular component analysis of nuclear RBPs identified by APEX-PS. **f** RBP specificity of nuclear datasets. For comparison, nuclear RBPs identified by fractionation[43, 44] were analyzed in the same manner. Details in Supplementary Data 2. **g** Nuclear specificity of nuclear RBP datasets identified by APEX-PS, compared to nuclear RBPs identified by fractionation (RBR-ID and serIC). **h** Using a list of 155 true-positive nuclear RBPs, the coverage of APEX-PS was compared to fractionation-based methods. **i** Overlap of APEX-PS datasets with global RBP dataset obtained by OOPS[26]. Bottom: Comparison of protein abundance for RBPs identified by both methods to those identified by APEX-PS only. **j** Subclassification of RBPs in nuclear APEX-PS dataset. Many RBPs we enriched bind to poly(A). Of the remainder, 90 have been experimentally shown to bind to the 7 RNA classes at right. Details in Supplementary Data 2. **k** Comparison of FA-dependent APEX-PS enrichment for RRM-containing RBPs (green) versus unknown (gray)-RBD-containing proteins in the nuclear APEX-PS dataset. Box limits represent 25th percentiles, medians, and 75th percentiles. Statistical analysis was performed with one-sided Wilcoxon rank sum test.

higher, as many of the "orphans" we identified (proteins lacking prior RBP annotation) may represent newly discovered nuclear RBPs. For instance, we enriched many orphan proteins related to transcription and chromatin modification, which may represent new RBPs involved in genome regulation (Supplementary Fig. 6b). The Venn diagram comparing the true-positive proteins identified by the three methods is shown in Fig. 3h and APEX-PS recovered the largest subset of these (72 proteins).

The nuclear RBPome identified by APEX-PS has extensive overlap with previous datasets generated by PS-based methods[26–28] (Supplementary Fig. 6c). Interestingly, APEX-PS identified 272 nuclear RBPs missed by OOPS; these tend to be lower-abundance proteins[58] (Fig. 3i). We surmise that by focusing on a single subcellular compartment, APEX-PS might probe more deeply than global profiling methods and identify lower-abundance RBPs. We also crossed our dataset with previous polyadenylated RBP datasets[20,21,26,43,55,56] and estimate that 50% of our RBPs bind to polyadenylated RNAs. Among the remaining nuclear RBPs, we found that some of them bind to other RNA classes, such as tRNA, rRNA, and snoRNA, consistent with the ability of PS to enrich all types of RBPs (Fig. 3j).

RBPs bind to RNA via modular RNA-binding domains (RBDs) that have been classified into 11 classical and 15 nonclassical subtypes[21,59]. In our nuclear RBPome, 115 and 121 nuclear RBPs contain classical and nonclassical RBDs, respectively (Supplementary Fig. 6d, e). Consistent with previous RBP studies, a large percentage of RBPs that we mapped do not have characterized RBDs, suggesting that they may bind RNA through novel mechanisms. We found that RBPs containing the RNA recognition motif (RRM), which interacts tightly (nanomolar affinity) with RNA-binding partners via multiple sequential stacking interactions[16,60], generally displayed low +FA/−FA enrichment ratios, suggesting that high-affinity RBPs can be enriched by phase separation even in the absence of chemical crosslinking (Fig. 3k), as has been reported previously[26].

**Identification of nucleolar RBPs by APEX-PS.** Having validated APEX-PS in the nucleus, we turned our attention to the nucleolus, a membraneless organelle that cannot be purified by biochemical fractionation. The nucleolus is the primary site for ribosome biogenesis and plays critical roles in many physiological and pathological processes, during which resident proteins interact extensively and dynamically with RNAs[61,62]. We performed three biological replicates of APEX-PS in HEK293T cells stably expressing APEX2 targeted to the nucleolus (Fig. 3a, b), observing excellent correlation across replicates (Supplementary Fig. 7a). As in previous studies[63,64], we utilized whole-nucleus APEX2-NLS samples as spatial reference in order to enhance

spatial specificity. We filtered the mass spectrometric data by both the "pairwise ROC approach" (Fig. 4a, b and Supplementary Fig. 7b) and the "statistical approach". The former yielded a nucleolar RBPome of 252 proteins (Supplementary Data 3). Of note, pairwise ROC analysis using different pairings of NIK3x and NLS samples had negligible impact on the composition of the final nucleolar RBPome (Supplementary Fig. 8a, b). The statistical filtering approach produced a smaller proteome of 223 nucleolar RBPs (Supplementary Fig. 8c, d and Supplementary Data 3). Due to superior sensitivity and comparable specificity (Supplementary Fig. 8e-g), we moved ahead with the pairwise ROC-generated nucleolar RBPome.

Gene ontology analysis showed significant enrichment of nucleolus (Fig. 4c) and expected biological process (Fig. 4d) terms, including rRNA processing and translation initiation, for our nucleolar RBP dataset. We found 29 ribosomal protein subunits in our dataset, in addition to several H/ACA and Box C/D small nucleolar ribonucleoprotein complexes (snoRNP), which catalyze the pseudouridylation and 2'-O-ribose methylation of rRNA, respectively[65] (Fig. 4e). Eighty one percent and 96% of our nucleolar proteins have prior nuclear and RNA-binding annotation, respectively (Fig. 4f, g). As an additional check of specificity, we examined the components of the nuclear pore complex (NPC), which is in the vicinity of nucleoli but not part of nucleoli. Whereas we enriched nine components of the NPC in our whole-nucleus APEX-PS dataset, none were enriched in our nucleolar APEX-PS dataset (Supplementary Fig. 8h). Comparing again to global OOPS[26], we found that 86 of our nucleolar RBPs were missed by OOPS, and that these are generally lower-abundance proteins (Fig. 4h). Similarly, 92 nucleolar RBPs were missed by poly (A)-dependent methods[20,21,26,43,55,56], perhaps because they bind to non-coding RNAs (Supplementary Fig. 8i).

Our nucleolar RBP dataset contains nine "orphan" proteins that have neither prior RNA binding nor nucleolar annotation. These could be nonspecific proteins, or they could be newly discovered nucleolus-localized RBPs. We selected two hits for follow-up validation: MIS18A (Protein Mis18-alpha) and ATAD5 (ATPase family AAA domain-containing protein 5). We first repeated APEX-PS using APEX-NIK3x cells, but used UV crosslinking instead of FA-based RNA–protein crosslinking. The western blot in Fig. 4i shows enrichment of both proteins by APEX-PS (UV) in the nucleolus. Second, we used an orthogonal RBP enrichment method consisting of RNA metabolic labeling followed by click chemistry and pulldown[24], which again enriched both proteins (Fig. 4j). Third, we performed fluorescence imaging and found that MIS18A is highly enriched in the nucleolus while ATAD5 is widely distributed in the nucleus, overlapping with a nucleolar marker (Fig. 4k). These data suggest

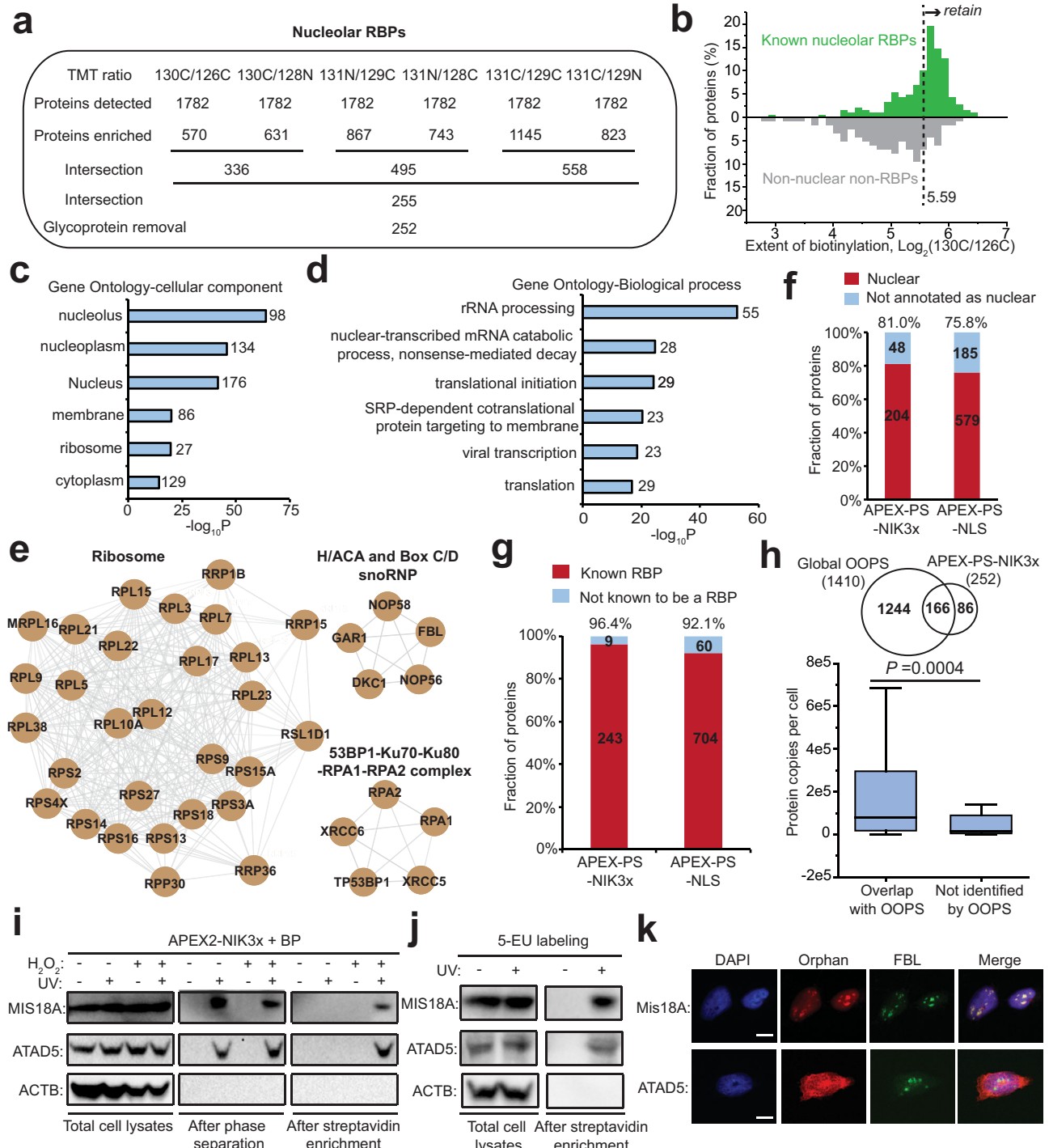

**Fig. 4 Proteomic profiling of nucleolar RBPs by APEX-PS. a** Numbers of proteins remaining after each step of filtering the mass spectrometric data using the "pairwise ROC strategy". The final nucleolar RBPome has 252 proteins (Supplementary Data 3). **b** Sample histogram showing how the cutoff for 130 C/126 C TMT ratio was applied. **c** GO cellular component analysis of nucleolar RBPs identified by APEX-PS. **d** GO biological process analysis of nucleolar RBPs identified by APEX-PS. **e** Molecular complexes enriched in nucleolar RBPome. Gray lines indicate protein-protein interactions annotated by the STRING database. **f** Nuclear specificity of nucleolar RBPome. **g** RBP specificity of nucleolar dataset. See Supplementary Data 3 for details. **h** Overlap with global RBP dataset obtained by OOPS[26]. Bottom: Comparison of protein abundance for RBPs identified by both methods to those identified by APEX-PS only. Box limits represent 25th percentiles, medians, and 75th percentiles. Statistical analysis was performed with one-sided Wilcoxon rank sum test. **i** Validation of MIS18A and ATAD5 as novel nucleolar RBPs by UV crosslinking APEX-PS. **j** Validation of the RNA binding of MIS18A and ATAD5 by metabolic labeling of RNA by 5-EU and UV crosslinking. **k** Confocal imaging of endogenous MIS18A and ATAD5. Anti-FBL stains nucleolus and DAPI stains nuclei. Scale bars, 10 μm.

that our nucleolar RBPome can be mined for novel nucleolus-localized RNA-binding proteins.

**Discovery of RBPs at the outer mitochondrial membrane by APEX-PS.** The landscape of RBPs at the mammalian outer mitochondrial membrane (OMM) has not previously been explored but could hold clues to the mechanisms of mitochondrial protein translation. Of the >1200 proteins assigned to the mitochondrion, only 13 are encoded by the mitochondrial genome. The remainder are encoded by the nuclear genome, and their protein products must be imported into the mitochondrion after translation in the cytosol. Since the detection of ribosomes at the OMM by electron microscopy[66], several studies[11,67], including ours, have detected mitochondrial mRNAs proximal to the OMM, suggesting that these may be translated "locally" to facilitate co-translational or post-translational import of their protein products in the mitochondrion. For instance, our APEX-seq study detected 1902 OMM-localized mRNAs in HEK cells[11], while proximity-specific ribosome profiling showed active translation of 551 mRNAs at the yeast OMM[11,67]. These observations raise the question of how mitochondrial mRNAs are recruited to the OMM for local translation—are specific OMM-localized RBPs involved?

To identify OMM-localized RBPs, HEK293T cells stably expressing APEX2-OMM were subjected to proximity biotinylation, FA crosslinking, and phase separation in two biological replicates, while cytosolic APEX2-NES was used as a spatial reference control in a TMT 11-plex experiment (Fig. 5a, Supplementary Fig. 9a, b and Supplementary Data 4). We also included two replicates of cells treated with puromycin (PUR), which inhibits protein translation and disassembles polysomes[68], in order to detect OMM-localized RBPs that bind to RNA in a translation- and ribosome-independent manner (Fig. 5b).

A total of 28 and 15 OMM-localized RBPs were identified under basal and PUR-treated conditions, respectively (Fig. 5c and Supplementary Data 5). The +PUR dataset is almost entirely a subset of the basal (−PUR) dataset (Fig. 5d). Most proteins in our OMM RBPome have prior mitochondrial annotation (~72%) and RNA-binding annotation (~76%) (Fig. 5e, f). Interestingly, several proteins (8 out of 28) also have literature connections to mitochondria-ER contact sites, which previous studies have linked to protein translation[64,69,70] (Supplementary Fig. 9c). Among the ribosome-independent OMM RBPs, only vimentin (VIM), an intermediate filament protein, exhibits increased RNA binding upon PUR-induced translation inhibition and might be involved in the regulation of released mRNAs (Supplementary Fig. 9d).

We selected five proteins for follow-up validation. TAX1BP1 (Tax1-binding protein 1), RMDN3 (Regulator of microtubule dynamics protein 3), and MARC1 (Mitochondrial amidoxime-reducing component 1) have not previously been shown to bind RNA and TAX1BP1 also lacks evidence of OMM localization. SYNJ2BP is a tail-anchored OMM protein with a cytosol-facing PDZ domain; we previously showed that the complex of SYNJ2BP and its ER binding partner, RRBP1, functions as a mitochondria-rough ER tether[64]. Although SYNJ2BP has been identified as an RBP by pCLAP[57], it was not detected in other RBP profiling studies and there have not been follow-up biochemical experiments investigating SYNJ2BP's RNA-binding functions. Another protein we selected, EXD2 (Exonuclease 3'-5' domain-containing protein 2), is an OMM-localized 3'-5' exonuclease that acts on single-stranded RNA[71] and has also been detected at mitochondria-ER contact sites[70,72]. Using UV-based APEX-PS and the metabolic labeling approach[24], all five of these proteins were verified as bona fide RBPs, whereas the

negative control ACTB was not enriched (Fig. 6a, b). The OMM localization of TAX1BP1 was also confirmed by immunofluorescence staining (Fig. 6c). As EXD2 localizes to both the OMM and nucleus, we performed APEX2-OMM and APEX2-NLS labeling in parallel to compare the RNA-binding activities of EXD2 in these two compartments. UV-based APEX-PS showed, surprisingly, that EXD2 preferentially binds to RNA at the OMM, highlighting the powerful capability of this methodology to dissect heterogeneity in RNA-binding functions of proteins localized to multiple subcellular regions (Fig. 6d).

**Analysis of the mRNA clients of the OMM-localized RBP SYNJ2BP.** We were intrigued by SYNJ2BP and especially its persistence as an RBP after puromycin treatment, suggesting that it binds to RNA in a ribosome- and translation-independent manner. To further probe the function of SYNJ2BP, we performed RNA immunoprecipitation and sequencing (RIP-seq) to identify the potential mRNA clients of SYNJ2BP. Of the >100 mRNAs enriched (Supplementary Data 6), we selected 11 mitochondrion-related hits with a range of fold-enrichment values for follow-up validation by CLIP (crosslinking immuno-precipitation), which is more specific than RIP. Figure 6e and Supplementary Fig. 10a show that all 11 mRNAs were enriched by CLIP-based pulldown of endogenous SYNJ2BP, while several negative control mRNAs were not enriched. These negative controls include IARS2 (Isoleucine-tRNA ligase, mitochondrial), ATP5O (ATP synthase subunit O, mitochondrial), and MRPS22 (28 S ribosomal protein S22, mitochondrial), all of which were enriched at the OMM by APEX-seq[11] but not by SYNJ2BP RIP-seq. We repeated the CLIP analysis following PUR treatment, and again found all 11 mRNA clients enriched to a similar extent, suggesting that SYNJ2BP binds to these mRNAs regardless of ribosome activity. Imaging confirmed that SYNJ2BP remains OMM localized after PUR treatment (Supplementary Fig. 10b).

To determine if these 11 mRNAs are localized to the OMM solely through the action of SYNJ2BP or if other binding interactions also play a role, we used OMM-localized APEX to directly biotinylate the RNAs at the OMM in both wild-type and SYNJ2BP knockout HEK293T cells (Supplementary Fig. 10c). We found that 5 of the 11 mRNAs were significantly reduced at the OMM when SYJN2BP was absent (Fig. 6f and Supplementary Fig. 10d). This effect was specific to the PUR-treated condition only. Our observations suggest that binding to SYNJ2BP (or SYNJ2BP-dependent binding to other RBPs) is a major mechanism for retention of a specific subset of mRNAs at the OMM following PUR treatment, but other mechanisms exist for recruiting these mRNAs to the OMM under basal conditions—for instance, interactions between the TOM mitochondrial protein import machinery and the ribosome–mRNA–nascent protein chain complex that displays an N-terminal mitochondrial targeting sequence[73]. The five SYNJ2BP-dependent mRNAs encode three oxidative phosphorylation (OXPHOS)-related proteins (UQCR11, PET117, and RAB5IF), a mitochondrial ribosome component (MRPS17) and a key mitochondrial fission factor (MTFP1).

**SYNJ2BP retains specific mitochondrial mRNAs at the OMM to facilitate restoration of mitochondrial function after stress.** Due to the functional importance of SYNJ2BP's five validated mRNA clients, we hypothesized that SYNJ2BP may retain them at the OMM, even through periods of stress, in order to facilitate their rapid local translation for restoration of mitochondrial function. To test this hypothesis, we examined the effect of SYNJ2BP KO on the levels of proteins encoded by these genes, under basal conditions, after PUR treatment, and after 12 h of

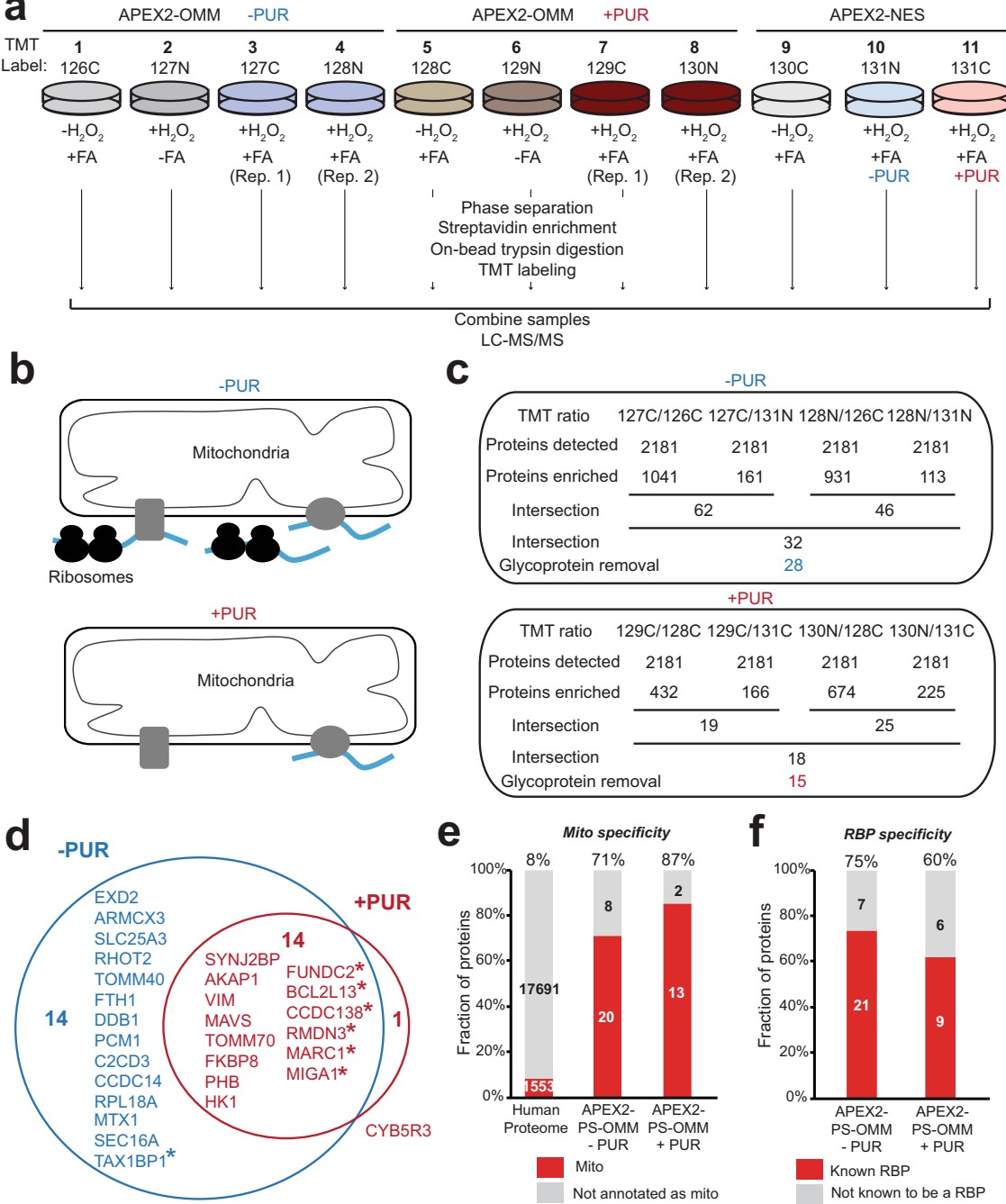

**Fig. 5 Proteomic profiling of RBPs localized to the outer mitochondrial membrane by APEX-PS. a** Experimental design and labeling conditions for TMT-based proteomics. HEK cells stably expressing APEX2-OMM were subjected to proximity biotinylation, FA crosslinking, and enrichment according to Fig. 1a-b. The control samples omitted $H_2O_2$ (samples 1 and 5) or FA (samples 2 and 6). Cytosolic APEX2-NES was included as a reference for ratiometric analysis (samples 9–11). **b** Schematic showing expected changes at the OMM upon treatment with puromycin (PUR), which inhibits protein translation and disrupts polysomes. Blue lines, RNA. Gray shapes, potential RBPs. **c** Numbers of proteins remaining after each step of filtering the mass spectrometric data. The final nuclear −PUR and +PUR OMM RBP proteomes obtained by APEX-PS have 28 and 15 proteins, respectively. **d** Overlap of OMM-localized RBPs under basal and +PUR conditions. RBP orphans (proteins with no previous RNA-binding annotation in literature) are starred. **e** Mitochondrial specificity of OMM RBPs identified by APEX-PS. For comparison, same analysis was performed on entire human proteome. **f** RBP specificity of APEX-PS OMM datasets. Details in Supplementary Data 5.

recovery from PUR treatment. No change in protein levels were observed under basal and PUR conditions, but all five proteins were reduced in abundance upon SYNJ2BP KO 12 h after recovery from PUR (Supplementary Fig. 11a). Because long protein half-lives may obscure the full effect of SYNJ2BP on protein translation, we repeated the assay but used azidohomoalanine

(AHA) (followed by Click reaction with alkyne-biotin and streptavidin-based enrichment[74]) to selectively detect newly synthesized proteins. Figure 6g shows that SYNJ2BP KO clearly impaired the translation of all five proteins during the PUR recovery phase. In a control experiment, SYNJ2BP had no impact on the total protein level or newly synthesized protein level of

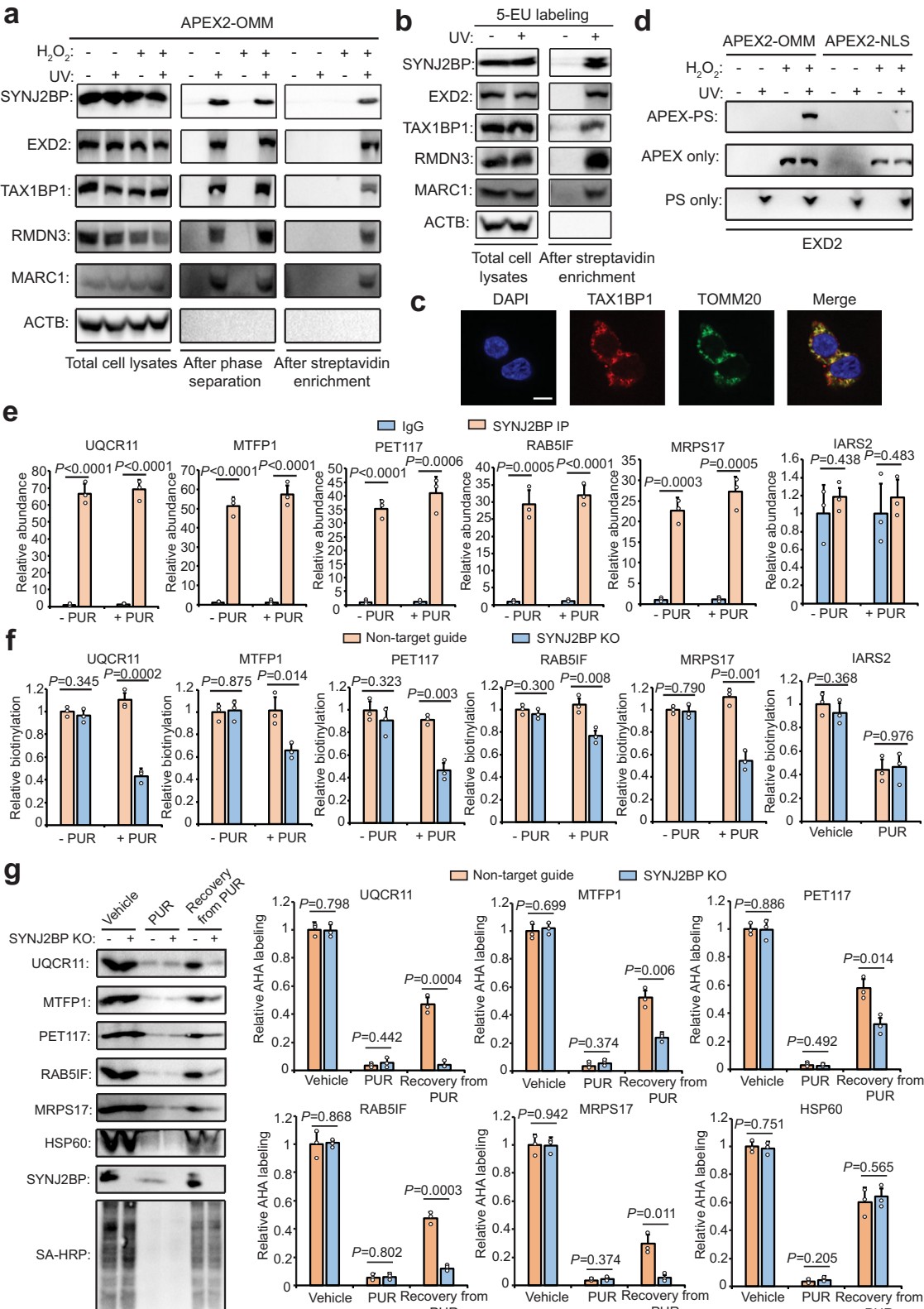

HSP60 (60 kDa mitochondrial heat shock protein), a mitochondrial protein whose mRNA is not a client of SYNJ2BP.

To investigate whether SYNJ2BP's role in mitochondrial protein translation has an effect on overall mitochondrial function, we first performed measurements of OXPHOS—specifically the activities of Complex III and Complex IV. One of SYNJ2BP's validated mRNA clients, UQCR11 (Cytochrome

b-c1 complex subunit 10), encodes a subunit of Complex III[75], which catalyzes electron transfer from ubiquinol to cytochrome c. Figure 7a shows that PUR treatment, unsurprisingly, impairs the activity of Complex III. During the recovery from PUR stress, the Complex III activity of wild-type HEK cells recovers to some extent, but the Complex III activity of SYNJ2BP KO cells fails to recover. We also measured the activity of Complex IV, whose

**Fig. 6 SYNJ2BP binds to specific mitochondrial mRNAs at the OMM and promotes their translation during recovery from stress. a** Validation of five hits from Fig. 5d as OMM-localized RBPs by UV crosslinking-based APEX-PS. Non-RBP ACTB was used as a negative control. **b** Validation of RNA-binding activity using metabolic RNA labeling by 5-EU and UV crosslinking. **c** Confocal imaging of mitochondrial RBP orphan TAX1BP1. Anti-TOMM20 stains mitochondria and DAPI stains nuclei. Scale bars, 10 μm. **d** Comparison of RNA-binding activities for OMM-localized and nuclear-localized EXD2 using APEX-PS-OMM and APEX-PS-NLS. Direct streptavidin enrichment (APEX only) was performed to quantify total EXD2 protein levels in each compartment. PS only condition shows RNA-binding activity of total EXD2. **e** Validation of mitochondria-related SYNJ2BP clients in the absence or presence of protein translation inhibitor PUR by CLIP and qRT-PCR. IARS2 is an OMM-localized mRNA not identified to bind with SYNJ2BP. Other SYNJ2BP clients and negative controls are shown in Supplementary Fig. 10a. **f** Evaluation of OMM localization of SYNJ2BP clients in SYNJ2BP knockout cells by APEX-mediated proximity labeling of RNA. The OMM localization was determined by comparing APEX2-OMM with APEX2-NES labeling (y-axis). Other SYNJ2BP clients and negative controls are shown in Supplementary Fig. 10d. **g** Impact of SYNJ2BP knockout on protein synthesis of SYNJ2BP-regulated clients under different cellular states. The newly synthesized proteins were labeled by AHA and captured by click-based enrichment. HSP60 is a mitochondrial protein not identified as a target of SYNJ2BP. Right: quantification of western blot data from three biological replicates. Two-sided Student's t-test was performed and values represent means ± SD from three biological replicates.

assembly requires PET117 (Protein PET117 homolog, mitochondrial), another validated SYNJ2BP client. Previous work has suggested that PET117 is essential for OXPHOS and PET117 mutation leads to neurodevelopmental regression due to a deficiency in Complex IV assembly[76,77]. Figure 7b shows PUR-induced decrease in Complex IV activity, and failure to return to previous levels of activity during stress recovery, only in SYNJ2BP KO cells but not in wild-type cells.

We also used an antibody cocktail to probe the assembly of Complexes I–V and observed disruption of Complex III and IV assembly in the absence of SYNJ2BP during cell recovery from PUR stress (Fig. 7c). This further supports the notion of specific regulation of OXPHOS by SYNJ2BP. The assembly of Complex I was also impaired to some extent, and may be mediated by other SYNJ2BP clients such as RAB5IF, which is a potential regulator of OXPHOS complexes[78].

To interrogate the role of SYNJ2BP in mitochondrial health overall, we performed measurements of cell proliferation, which depends on mitochondrial activity, especially when cells are cultured in galactose. Cells grown in glucose derive ATP from both aerobic glycolysis and mitochondrial glutamine oxidation, whereas cells growth in galactose derive a larger fraction of their ATP from mitochondrial respiration[79]. We found that in glucose media, WT and SYNJ2BP KO cells grew identically under basal conditions, but SYNJ2BP KO cells grew more slowly than wild type after PUR treatment (Fig. 7d and Supplementary Fig. 11b). This effect was not observed using cycloheximide (CHX), which also inhibits protein translation but keeps polysomes intact (Supplementary Fig. 11c). In the presence of CHX, the OMM-localized mRNAs could be retained by polysomes and thus cells survived in a SYNJ2BP-independent manner. When the cells were grown in galactose to increase reliance on mitochondrial respiration, the effect of SYNJ2BP KO on cell growth following PUR treatment became more pronounced (Supplementary Fig. 11d). We also measured ATP levels directly and found that SYNJ2BP KO decreased ATP in galactose-cultured cells after PUR treatment, but not under basal or CHX-treated conditions (Supplementary Fig. 11e). Collectively, our results suggest that SYNJ2BP retains specific mitochondrial transcripts at the OMM following PUR-specific translation inhibition stress, and that this serves to improve the recovery of OXPHOS, mitochondrial ATP production, and cell growth from such stress.

Apart from PUR treatment, other types of stress, such as heat and oxidative stress, can also suppress translation and dissociate mRNAs from polysomes[80]. The released mRNAs interact extensively with RBPs, inducing the formation of molecular condensates such as stress granules[81]. To test if SYNJ2BP also plays a role in the recovery of mitochondrial function from other types of translation stress, we first used heat or sodium arsenite to stress cells, and then checked for SYNJ2BP-dependent retention

of five mRNA clients by OMM-targeted APEX RNA labeling. Supplementary Figure 12a shows that all five mRNAs associate with the mitochondrion under heat and sodium arsenite stress in wide-type cells, but largely disappear from the OMM in SYNJ2BP knockout cells. Moreover, loss of SYNJ2BP also inhibits the translation of these mRNAs under both stresses, which further leads to the inhibition of Complexes III and IV (Fig. 7e-g and Supplementary Fig. 12b-f). In a proliferation assay, SYNJ2BP knockout impaired cell growth during recovery from heat and sodium arsenite stresses (Supplementary Fig. S12g-h). Taken together, SYNJ2BP may represent a novel mechanism for the specific regulation of local translation at the OMM in response to cellular stresses (Fig. 7h).

## Discussion

Proximity labeling (PL) with enzymes, such as APEX, TurboID, and BioID, has been widely applied and is straightforward to implement. Here, we show that the simple combination of PL with functional protein enrichment workflows produces datasets that have functional annotation in addition to the usual spatio-temporal assignments that PL methods provide. We found that the approach is very versatile: APEX-based PL can be combined with IMAC enrichment to resolve spatial phosphoproteomes, with WGA affinity chromatography to resolve spatial O-GlcNAcylated proteomes, and with phase separation to resolve spatial RBPomes. Functional PL may also be useful in other ways not explored here. The methods should be straightforwardly extensible to other unpurifiable compartments, especially those already mapped by PL techniques including stress granules[82], the ER membrane[64], and lipid droplets[83]. In vivo organ-specific, cell-type specific, or organelle-specific functional PL may also be possible, especially if APEX is replaced by TurboID.

RBPs constitute an important subclass of the human proteome, with essential roles in transcription, translation, chromatin organization, and stress response[16,18,41]. Here we showed that APEX-PS can be used for unbiased discovery of RBPs in specific subcellular compartments. In the nucleus, APEX-PS identified RBPs with high specificity and sensitivity compared to previous fractionation-based methods. We also showed that APEX-PS can be applied for RBP discovery in unpurifiable subcompartments, such as the nucleolus and outer mitochondrial membrane (OMM), which are inaccessible to biochemical fractionation. Taking advantage of the ~11 min temporal resolution of APEX-PS, we compared the RBPome of the OMM before and after translation inhibition stress.

Many innovative strategies for large-scale discovery of RBPs have been reported in recent years. UV/formaldehyde cross-linking followed by oligo(dT) bead capture can enrich RBPs that bind to polyadenylated RNAs[20–23]. Metabolic labeling of RNA with alkyne analogs used in CARIC[24] and RICK[25] identifies

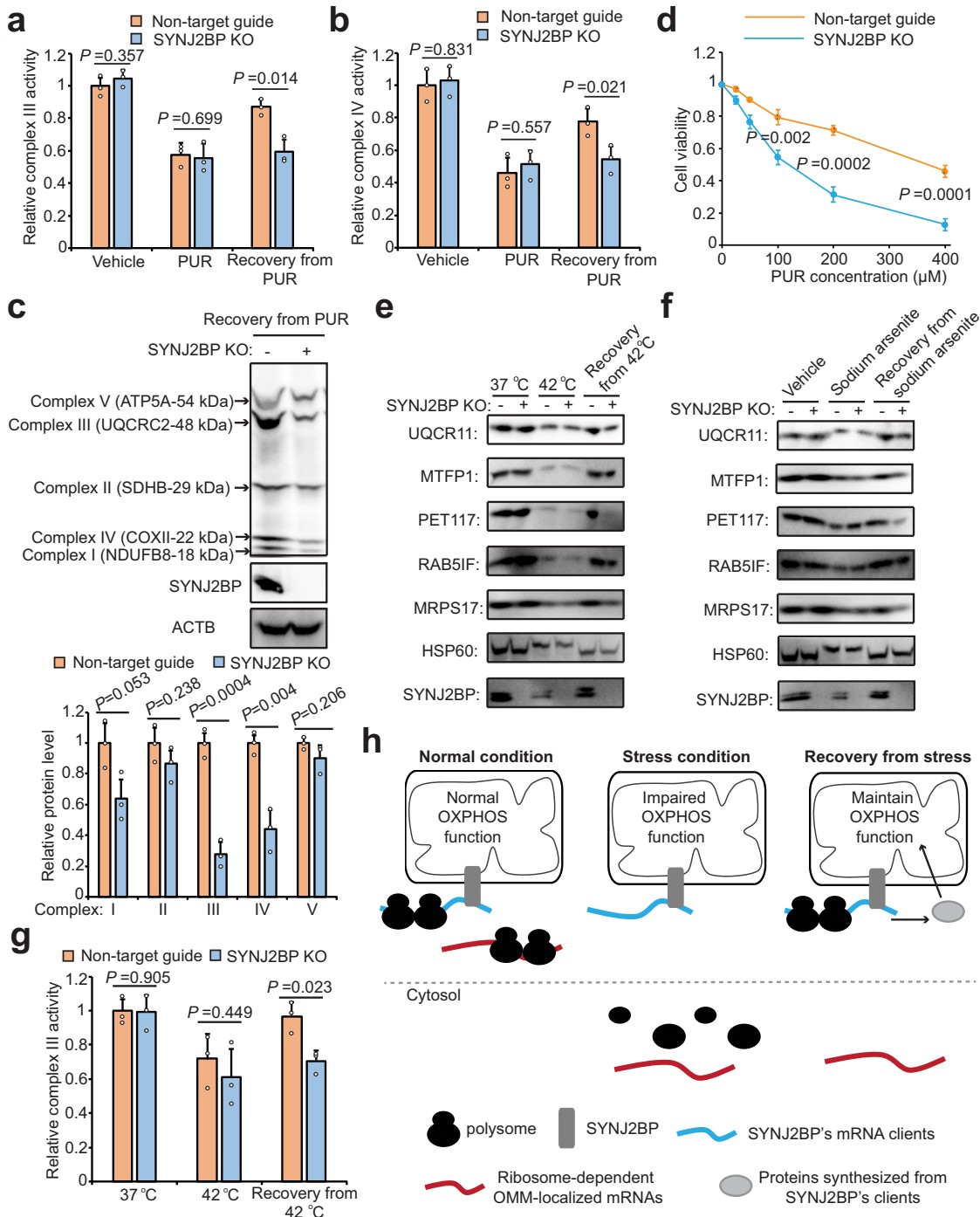

**Fig. 7 SYNJ2BP improves mitochondrial OXPHOS recovery and cell viability following stress. a**, **b** Evaluation of Complex III (**a**) and IV (**b**) activities in SYNJ2BP knockout cells. **c** Evaluation of complex I–V assembly 12 h following treatment of HEK cells with protein translation inhibitor PUR. The quantification of each OXPHOS complex is shown below; data from three independent biological replicates. **d** Evaluation of cell viability 12 h following treatment of HEK cells with PUR. **e**, **f** Evaluation of protein synthesis for SYNJ2BP clients in SYNJ2BP knockout cells following heat (**e**) and sodium arsenite (**f**) stresses. The quantification of western blot data from three biological replicates is shown in Supplementary Fig. 12b-c. **g** Evaluation of Complex III activity in SYNJ2BP knockout cells following heat stress. **h** Working model of SYNJ2BP's role in retaining important (blue) mRNAs at the OMM during stress, to facilitate their rapid local translation for restoration of mitochondrial function during stress recovery (right). Two-sided Student's *t*-test was performed and values represent means ± SD from three biological replicates.

polyadenylation-independent RBPs. Several groups have reported organic-aqueous phase separation for RBP enrichment independent of RNA class[26–28]. While these methods have uncovered thousands of newly annotated RBPs, the vast majority of them have unknown or partially known function. As a first step in elucidating their biology, the spatial assignment of RBPs to

specific subcellular locales, under specific cellular states, could help to shed light on their functional roles.

To recover spatial information, previous studies have crossed RBP datasets with protein localization datasets[84], even though such datasets are often acquired in different cell types under different conditions. Moreover, many RBPs reside in multiple

subcellular locations but preferentially bind to RNA in one of these locations (for example, EXD2, as demonstrated above); such information is lost by dataset crossing. Another, more direct, approach to spatial assignment of RBPs is to combine RBP profiling methods with subcellular fractionation. This is effective for organelles that can be enriched by biochemical fractionation, such as the nucleus[85], but inapplicable to the many subcellular regions that are impossible to purify, such as stress granules, processing bodies, and the nucleolus and OMM that we map here with APEX-PS.

We used APEX-PS to investigate the biology of the outer mitochondrial membrane (OMM) and identified 15 RBPs at the OMM following PUR treatment. These proteins are candidates for recruiting and/or retaining mitochondrial mRNAs at the OMM for local protein translation. We examined one of our OMM RBP hits in-depth—SYNJ2BP—and discovered that it safeguards the OMM localization of its mRNA clients upon disassembly of polysomes during stress, and facilitates the translation of these mRNAs during mitochondrial recovery from stress. As some OXPHOS-related mRNAs are clients of SYNJ2BP, we found that loss of SYNJ2BP impaired the activities of Complexes III and IV, as well as overall cell viability, during the stress recovery process. A working model is that SYNJ2BP binds to a subset of mitochondrial mRNAs and regulates their translation in response to inputs such as cell stress, ER status (due to SYNJ2BP's role as a mitochondria-ER tether[64]), and transcriptional events.

## Methods

**Cell culture**. HEK293T cells from the ATCC (passages < 25) were cultured in a 1:1 DMEM/MEM mixture (Cellgro) supplemented with 10% fetal bovine serum, 100 units/mL penicillin, and 100 mg/mL streptomycin at 37 °C under 5% $CO_2$. For fluorescence microscopy imaging experiments, cells were grown on $7 \times 7$ mm glass coverslips in 48-well plates. For APEX-PS, APEX-IMAC and APEX-WGA experiments, cells were grown on 15 cm glass-bottomed Petri dishes (Corning). To improve the adherence of HEK293T cells, glass slides and plates were pretreated with 50 mg/mL fibronectin (Millipore) for 20 min at 37 °C before cell plating and washing three times with Dulbecco's PBS (DPBS) (pH 7.4). HEK293T cells stably expressing APEX2-NLS, APEX2-NIK3x, APEX2-OMM, APEX2-NES, and ERM-APEX2 were generated in our previous studies[11].

**APEX labeling**. For both western blot analysis and proteomic analysis, HEK293T cells stably expressing the APEX2 fusion construct of interest were cultured in 15 cm dish for 18–24 h to about 90% confluency. APEX labeling was initiated by changing to fresh medium containing 500 µM biotin-phenol (cat.no. A8011, ApexBio) and incubating at 37 °C under 5% $CO_2$ for 30 min. $H_2O_2$ (Sigma–Aldrich) was then added to a final concentration of 1 mM and the plate was gently agitated for 1 min. For OMM-localized RBP profiling upon PUR treatment, HEK293T cells stably expressing APEX2-OMM were treated with 200 µM puromycin and 500 µM biotin-phenol for 30 min and then APEX labeling was initiated by $H_2O_2$ treatment for 1 min.

For functional PL to enrich subcellular phosphorylated and O-GlcNAcylated proteins, the media of $H_2O_2$-treated cells was aspirated and the APEX labeling reaction was quenched by addition of 10 mL quenching solution (10 mM ascorbate, 5 mM Trolox, and 10 mM sodium azide in DPBS). Cells were washed with quenching solution for three times and media were aspirated. Cells were freshly subjected to enrichment of O-GlcNAcylation and phosphorylation described below.

For FA-based APEX-PS, media was aspirated and the APEX labeling reaction was quenched by addition of 10 mL azide-free quenching solution (10 mM ascorbate and 5 mM Trolox in DPBS). Cells were incubated at room temperature for 1 min, then media was removed by aspiration and 10 mL of crosslink-quench solution (0.1% (v/v) formaldehyde, 10 mM sodium ascorbate and 5 mM Trolox in DPBS) was added. After 1 min, media were aspirated and cells were again incubated in 10 mL fresh crosslink-quench solution for 9 min at room temperature with gentle agitation. The crosslinking reaction was terminated in 125 mM of glycine for 5 min at room temperature. Cells were washed twice with 10 mL room-temperature DPBS, harvested by scraping, pelleted by centrifugation, and either processed immediately or flash frozen in liquid nitrogen and stored at −80 °C for further analysis.

For UV-based APEX-PS, the reaction was quenched by replacing the medium with an equal volume of quenching solution (10 mM ascorbate, 5 mM Trolox and 10 mM sodium azide in DPBS). Cells were washed with quenching solution for three times and media were aspirated. UV crosslinking was performed on PBS-washed cells by UV irradiation at 254 nm with 400 mJ/cm² (CL-1000 Ultraviolet

Crosslinker, UVP). Cells were washed twice with 10 mL ice-cold DPBS, harvested by scraping, pelleted by centrifugation, and either processed immediately or flash frozen in liquid nitrogen and stored at −80 °C before further analysis.

**IMAC enrichment of phosphoproteins**. The enrichment of phosphoproteins was performed by Pierce Phosphoprotein Enrichment Kit with phospho-specific IMAC affinity resins (cat.no.90003, ThermoFisher Scientific). The 15 cm cell culture plate of APEX labeled cells (~5 × 10⁷ cells) was washed with 10 mL nonphosphate buffer (50 mM HEPES, pH 7.0) twice. Cells were lysed by adding 1 mL of Lysis/Binding/Wash Buffer with CHAPS, 1× protease inhibitor cocktail (Sigma–Aldrich) to each cell culture plate. Cells were scraped and incubated on ice for 45 min with vortexing periodically. Cellular debris was removed by centrifugation at 10,000 × g for 20 min at 4 °C and the supernatant was collected. Protein concentration was determined by Pierce BCA protein assay kit (cat.no.23225, ThermoFisher Scientific). Approximately 4 mg of total cell extract was diluted to 0.5 mg/mL in Lysis/Binding/Wash Buffer. The diluted cell extract was added to a pre-equilibrated Phosphoprotein Enrichment Column and incubated for 30 min on a platform rocker at 4 °C. The Phosphoprotein Enrichment Column was washed with 5 mL Lysis/Binding/Wash Buffer for three times. Phosphoproteins were eluted by 1 mL Elution Buffer with agitation at r.t. for 3 min and the elution was performed for five times in total. Elution fractions were pooled and concentrated to 200 µL by the Pierce concentrators. The eluates were then subjected to Amicon Ultra-0.5 mL centrifugal filters and washed with RIPA buffer (50 mM Tris pH 8.0, 150 mM NaCl, 0.1% SDS, 0.5% sodium deoxycholate, 1% Triton X-100, 1× protease inhibitor cocktail, and 1 mM PMSF) for three times. The proteins were resuspended in 1 mL RIPA buffer and subjected to the streptavidin-based enrichment described below.

**WGA enrichment of O-GlcNAcylated proteins**. The enrichment of O-GlcNAcylated proteins was performed by Glycoprotein isolation kit, WGA (cat. no.89805, ThermoFisher Scientific). To decrease the O-GlcNAc level, cells were treated with 100 µM OSMI-1 (cat.no. ab235455, abcam) for 24 h. APEX labeled cells from a 15 cm petri dish (~5 × 10⁷ cells) were scraped, washed with PBS twice and lysed in 800 µL 1× Binding/Wash Buffer with sonication. Cellular debris was removed by centrifugation at 10,000 × g for 20 min at 4 °C and the supernatant was collected. Protein concentration was determined by Pierce BCA protein assay kit. Two aliquots of 2 mg proteins were diluted to 2.5 mg/mL in 1× Binding/Wash Buffer and incubated within a pre-equilibrated WGA column for 10 min at r.t. with end-over-end mixing. The column was then washed with 400 µL 1× Binding/Wash Buffer for three times with 5 min incubation of each wash. Glycoproteins were eluted by 200 µL Elution Buffer with rotation at r.t. for 10 min and the elution step was repeated. The eluates from two aliquots were combined and then subjected to Amicon Ultra-0.5 mL centrifugal filters and washed with RIPA buffer for three times. The proteins were resuspended in 1 mL RIPA buffer and subjected to the streptavidin-based enrichment described below.

**Phase separation**. Cell pellets from a 15 cm petri dish (~5 × 10⁷ cells) were resuspended in 1 mL Trizol and the homogenized lysate was transferred to a new tube. After incubating at room temperature for 5 min to dissociate, 200 µL of chloroform (Fisher Scientific) were added and vortexed, and the sample was centrifuged for 15 min at 12,000 × g at 4 °C. The upper, aqueous phase (containing noncrosslinked RNAs) and the lower, organic phase (containing noncrosslinked proteins) was removed. Interface (containing the protein–RNA complexes) was resuspended in 1 mL Trizol and subjected to two more cycles of phase separation. Finally, the interface was precipitated by 1 mL methanol and pelleted by centrifugation at 14,000 × g, room temperature for 10 min. The precipitated proteins was resuspended in 100 µL of 100 mM triethylammonium bicarbonate (TEAB), 1 mM MgCl₂, 1% SDS and incubated at 95 °C for 20 min. The samples were cooled at room temperature for 5 min and digested with 2 µg RNase A, T1 mix (2 mg/mL of RNase A and 5000 U/mL of RNase T1, ThermoFisher Scientific) for 2 h at 37 °C. Another 2 µg of RNase mix was added and incubated overnight at 37 °C. The resulting solution was subjected to the final round of phase separation and the RBPs in organic phase was recovered by precipitation in 4.5 mL methanol with centrifugation at 14,000 × g, room temperature for 20 min. The protein pellets were resuspended in 1 mL methanol with sonication using a Misonix sonicator (0.5 s on, 0.5 s off, for a total of 10 s on), transferred to a new 1.5 mL Eppendorf tube and pelleted by centrifugation at 14,000 × g, room temperature for 10 min. The protein pellets were resuspended in 1 mL RIPA buffer (50 mM Tris pH 8.0, 150 mM NaCl, 0.1% SDS, 0.5% sodium deoxycholate, 1% Triton X-100, 1× protease inhibitor cocktail (Sigma–Aldrich), and 1 mM PMSF) with sonication (0.5 s on, 0.5 s off, for a total of 10 s on). The total RBP solution was then subjected for western blotting or streptavidin enrichment.

**Streptavidin bead-based enrichment**. To enrich biotinylated material from the total RBP solution, 300 µL streptavidin-coated magnetic beads (Pierce) were washed twice with RIPA buffer, then incubated with the 1 mL total RBP solution with rotation for 2 h at room temperature. The beads were subsequently washed twice with 1 mL of RIPA lysis buffer, once with 1 mL of 1 M KCl, once with 1 mL of 0.1 M Na₂CO₃, once with 1 mL of 2 M urea in 10 mM Tris-HCl (pH 8.0), and twice

with 1 mL RIPA lysis buffer. For western blotting analysis, the enriched proteins were eluted by boiling the beads in 75 μL of 3× protein loading buffer supplemented with 20 mM DTT and 2 mM biotin. For proteomic analysis, the beads were then resuspended in 1 mL fresh RIPA lysis buffer and transferred to a new Eppendorf tube. The beads were then washed with 1 mL wash buffer (75 mM NaCl in 50 mM Tris-HCl pH 8.0) twice. The beads were resuspended in 50 μL of wash buffer and shipped to Steve Carr's laboratory (Broad Institute) on dry ice for further processing and preparation for LC-MS/MS analysis.

**Western blots**. For all western blots, after SDS-PAGE, the gels were transferred to nitrocellulose membrane, and then stained by Ponceau S (5 min in 0.1% (w/v) Ponceau S in 5% acetic acid/water). The blots were then blocked in 5% (w/v) milk (LabScientific) in TBS-T (Tris-buffered saline, 0.1% Tween-20) for at least 30 min at room temperature. For streptavidin blotting, the blots were stained with 0.3 μg/mL streptavidin-HRP in 3% BSA (w/v) in TBS-T for 1 h at 4 °C. The blots were washed three times with TBS-T for 5 min each time before to development. For the specificity validation of APEX-IMAC in Fig. 1c, the blots were stained with primary antibodies in 3% BSA (w/v) in TBS-T for 2 h in room temperature or overnight at 4 °C. The primary antibodies include anti-SMAD2 (1:2500 dilution, ab63576, Abcam), anti-Histone H3 (1:5000 dilution, ab18521, Abcam), anti-TOMM20 (1:1000 dilution, sc-17764, Santa Cruz Biotechnology), and anti-CANX (1:5000 dilution, ab22595, Abcam). For the specificity validation of APEX-WGA in Fig. 1d, the blots were stained with anti-SP1 (1:1000 dilution, sc-17824, Santa Cruz Biotechnology), anti-FBL (1:2500 dilution, ab166630, Abcam), anti-SDHA (1:2500 dilution, ab137040, Abcam), and anti-NHP2L1 (1:2500 dilution, ab181982, Abcam) in 3% BSA (w/v) in TBS-T for 2 h in room temperature or overnight at 4 °C. For the detection of phosphorylated ERK2 in Fig. 1f, the blots were stained with anti-ERK2 (1:2500 dilution, ab32081, Abcam) in 3% BSA (w/v) in TBS-T for 2 h in room temperature or overnight at 4 °C. For the analysis of global phosphorylation and O-GlcNAcylation level in Supplementary Fig. 1b-c, the blots were stained with the pan-phospho-Serine/Threonine antibody (1:1000 dilution, AP0893, ABclonal), pan-phospho-tyrosine (1:1000 dilution, 8594 S, Cell Signaling Technology) and anti-O-GlcNAc RL2 antibody (1:1000 dilution, MABS157, EMD Millipore). For validation of the specificity of APEX-PS in Fig. 2d and Supplementary Fig. 2c, f, g, the blots were stained with anti-SRSF1 (1:2500 dilution, ab129108, Abcam), anti-hnRNPC (1:2500 dilution, ab133607, abcam), anti-GRSF1 (1:2500 dilution, ab205531, Abcam), and anti-ETS2 (1:1000 dilution, sc-365666, Santa Cruz Biotechnology) in 3% BSA (w/v) in TBS-T for 2 h in room temperature or overnight at 4 °C. For the comparison of UV and FA crosslinking in Supplementary Fig. 2e, the antibodies for SRSF1, hnRNPC, ETS2 and Histone H3 were used as described above. Other antibodies include anti-beta Actin-HRP (1:10000 dilution, MA5-15739-HRP, ThermoFisher Scientific) and anti-beta Tubulin-HRP (1:10000 dilution, MA5-16308-HRP, ThermoFisher Scientific). For validation of nucleolar RBPs in Fig. 4i, j, the blots were stained with anti-MIS18A (1:2000 dilution, PA5-54238, ThermoFisher Scientific) and anti-ATAD5 (1:1000 dilution, SAB4301194-100UL, Sigma–Aldrich) in 3% BSA (w/v) in TBS-T for 2 h in room temperature or overnight at 4 °C.

For validation of OMM-localized RBPs in Fig. 6a, b, d, the blots were stained with anti-SYNJ2BP (1:2000 dilution, HPA000866-100UL, Sigma–Aldrich), anti-EXD2 (1:2000 dilution, HPA005848-100UL, Sigma–Aldrich), anti-TAX1BP1 (1:1000 dilution, HPA024432-100UL, Sigma–Aldrich), anti-RMDN3 (1:1000 dilution, HPA009975-100UL, Sigma–Aldrich), and MARC1 (1:1000 dilution, ab198692, abcam) in 3% BSA (w/v) in TBS-T for 2 h in room temperature or overnight at 4 °C. For the evaluation of protein synthesis in Fig. 6g, Fig. 7e, f and Supplementary Fig. 11a, the blots were stained with anti-UQCR11 (1:1000 dilution, MBS715423, MyBioSource), anti-MTFP1 (1:2500 dilution, ab198217, Abcam), anti-PET117 (1:2500 dilution, PA5-61574, ThermoFisher Scientific), anti-RAB5IF (1:1000 dilution, PA543332, ThermoFisher Scientific), anti-MRPS17 (1:2500 dilution, ab175207, Abcam), and anti-HSP60 (1:2500 dilution, ab190828, Abcam) in TBS-T for 2 h in room temperature or overnight at 4 °C. For evaluating the assembly of OXPHOS complexes in Fig. 7c, the blots were stained with the total OXPHOS human antibody cocktail (1:1000 dilution, ab110411, abcam). After washing three times with TBS-T for 5 min each, the blots were stained with secondary antibodies in 3% BSA (w/v) in TBS-T for 2 h in room temperature. The blots were washed three times with TBS-T for 5 min each time before to development with Clarity Western ECL Blotting Substrates (Bio-Rad) and imaging on the ChemiDoc XRS + System (Bio-Rad).

**On-bead trypsin digestion of biotinylated proteins**. To prepare samples for mass spectrometry analysis, proteins bound to streptavidin beads were washed twice with 200 μL of 50 mM Tris-HCl buffer (pH 7.5) followed by two washes with 2 M urea/50 mM Tris (pH 7.5) buffer. The final volume of 2 M urea/50 mM Tris buffer (pH 7.5) was removed and beads were incubated with 50 μL of /50 mM Tris and 0.5 μg trypsin for 30 mins at 37 °C with shaking. After 30 mins, the supernatant was removed and transferred to a fresh tube containing LysC digest. The streptavidin beads were washed once with 50 μL of 50 mM Tris buffer (pH 7.5) and the wash was combined with the on-bead digest supernatant and digested on shaker for at least 3 h at 37 degrees. The eluate was reduced with 4 mM DTT for 30 min at 25 °C with shaking. The samples were alkylated with 10 mM iodoacetamide for 45 min in the dark at 25 °C with shaking. Then 0.5 μg of trypsin was added to the

sample and the digestion continued overnight at 25 °C with shaking. After digestion, samples were acidified (to pH < 3.0) by adding formic acid such that the sample contained ~1% formic acid. Samples were desalted on C18 StageTips and evaporated to near dryness in a vacuum concentrator.

**TMT labeling and fractionation of peptides**. Desalted peptides were labeled with TMT (11-plex) reagents[86]. Peptides were reconstituted in 100 μL of 50 mM HEPES. Each 0.8 mg vial of TMT reagent was reconstituted in 41 μL of anhydrous acetonitrile and added to the corresponding peptide sample for 1 h at room temperature. Labeling of samples with TMT reagents was completed with the design shown in Fig. 3a and Fig. 5a. TMT labeling reactions were quenched with 8 μL of 5% hydroxylamine at room temperature for 15 min with shaking, evaporated to dryness in a vacuum concentrator, and desalted on C18 StageTips. For each TMT 11-plex cassette, 50% of the sample was fractionated by basic pH reversed phase using StageTips while the other 50% of each sample was reserved for LC-MS analysis by a single-shot, long gradient. One StageTip was prepared per sample using 2 plugs of Styrene Divinylbenzene (SDB) (3 M) material. The StageTips were conditioned two times with 50 μL of 100% methanol, followed by 50 μL of 50% MeCN/0.1% formic acid, and two times with 50 μL of 0.1% formic acid. Sample was resuspended in 100 μL of 0.1% formic acid and loaded onto the stageTips and washed with 100 μL of 0.1% formic acid. Following this, sample was washed with 60 μL of 20 mM NH₄HCO₃/2% MeCN, this wash was saved and added to fraction 1. Next, sample was eluted from StageTip using the following concentrations of MeCN in 20 mM NH₄HCO₃: 10, 15, 20, 25, 30, 40, and 50%. For a total of six fractions, 10 and 40% (fractions 2 and 7) elutions were combined, as well as 15 and 50% elutions (fractions 3 and 8). The six fractions were dried by vacuum centrifugation.

**Liquid chromatography and mass spectrometry**. Fractionated peptides were resuspended in 8 μL of 0.1% formic acid and were analyzed by online nanoflow liquid chromatography-tandem mass spectrometry (LC-MS/MS) using a Q-Exactive Plus Orbitrap MS (ThermoFisher Scientific) coupled online to an Easy-nLC 1200 (ThermoFisher Scientific). Four microliters of each sample was loaded onto a microcapillary column (360 μm outer diameter × 75 μm inner diameter) containing an integrated electrospray emitter tip (10 μm), packed to ~20 cm with ReproSil-Pur C18-AQ 1.9 μm beads (Dr. Maisch GmbH) and heated to 50 °C. The HPLC solvent A was 3% MeCN, 0.1% formic acid, and the solvent B was 90% MeCN, 0.1% formic acid. The SDB fractions were measured using a 110 min MS method, which used the following gradient profile at 200 nL/min: (min:%B) 0:2; 1:6; 85:30; 94:60; 95:90; 100:90; 101:50; 110:50 (the last two steps at 500 nL/min flow rate). The Q-Exactive Plus Orbitrap MS was operated in the data-dependent acquisition mode acquiring HCD MS/MS scans (resolution = 35,000, quadrupole isolation width of 0.7 Da) after each MS1 scan (resolution = 70,000, 300–1800 m/z scan range) on the 12 most abundant ions using an MS1 target of $3 \times 10^6$ and an MS2 target of $5 \times 10^4$. The maximum ion time utilized for MS/MS scans was 120 ms and the HCD normalized collision energy was set to 30. The dynamic exclusion time was set to 20 s, and the peptide match and isotope exclusion functions were enabled. Charge exclusion was enabled for charge states that were unassigned, 1 and >6.

**Mass spectrometry data processing**. LC-MS/MS data collection was achieved by Xcalibur software v4.3 (ThermoFisher Scientific). Collected data were analyzed using Spectrum Mill software package v6.1pre-release (Agilent Technologies). Nearby MS scans with the similar precursor m/z were merged if they were within ±60 s retention time and ±1.4 m/z tolerance. MS/MS spectra were excluded from searching if they failed the quality filter by not having a precursor MH+ in the range of 750–4000. All extracted spectra were searched against a UniProt database (12/28/2017 containing human reference proteome sequences, common laboratory contaminants, and mycoplasma ribosomes). Search parameters included: parent and fragment mass tolerance of 20 p.p.m., 50% minimum matched peak intensity, and calculate reversed database scores enabled. The digestion enzyme search parameter used was Trypsin Allow P, which allows K-P and R-P cleavages. The missed cleavage allowance was set to 3. TMT labeling was required at lysine, but peptide N termini were allowed to be either labeled or unlabeled. Allowed variable modifications were protein N-terminal acetylation, pyro-glutamic acid, deamidated N, and oxidized methionine. Individual spectra were automatically assigned a confidence score using the Spectrum Mill autovalidation module. Score at the peptide mode was based on target-decoy false discovery rate (FDR) of 1%. Protein polishing autovalidation was then applied using an auto thresholding strategy. Relative abundances of proteins were determined using TMT reporter ion intensity ratios from each MS/MS spectrum and the mean ratio is calculated from all MS/MS spectra contributing to a protein subgroup. Proteins identified by 2 or more distinct peptides and a protein score of at least 20 were considered for the dataset.

**Analysis of proteomic data for nucleus and nucleolus**. To determine the cutoff in each biological replicate, we adopted the ratiometric analysis as previously described[63]. The original identified proteins are shown in Supplementary Data 1. For the assignment of nuclear RBPs, a list of gold standard nuclear RBPs were manually collected according to previous literature[17], which are listed in

Supplementary Data S1, tab 2. For the false-positives (FPs), a list of 588 mito-chondrial matrix proteins identified by previous APEX profiling[3] was applied (Supplementary Data S1, tab 2). For each replicate, the proteins were first ranked in a descending order according to the TMT ratio (128 N/126 C, 128 C/127 N, 129 N/127 N). Here we did not include −FA control for ROC analysis because tight-binding RBPs that remain bound to their RNA partners through cell lysis and phase separation were recovered even in the absence of FA crosslinking. Instead, the FA controls were used to assess the RNA-binding affinity of RBPs, such as for the RRM-containing proteins (Fig. 3k). For each protein on the ranked list, the accumulated true-positive count and false-positive count above its TMT ratio were calculated. A receiver operating characteristic (ROC) curve was plotted accordingly for each replicate (Supplementary Fig. 4c). The cutoff was set where true-positive rate–false-positive rate (TPR–FPR) maximized. Post-cutoff proteomic lists of the three biological replicates were intersected and proteins enriched in at least two biological replicates were collected. The potential glycosylated proteins were removed according to the annotation of glycoproteins or locations exclusively in the secretory pathway (e.g., ER/Golgi lumen, plasma membrane, extracellular regions) to obtain the nuclear RBPome1 list (Supplementary Data 2). For the statistical analysis of nuclear RBPs, we performed a moderated T-test (limma R package v4.1) to determine proteins significantly enriched in the experimental conditions compared to the negative enrichment controls (omitting either APEX2 or $H_2O_2$). We corrected for multiple hypotheses (Benjamini–Hochberg procedure) and any proteins with an adjusted p-value of less than 0.05 were considered sta-tistically enriched. The potential glycosylated proteins were then removed to obtain the nuclear RBPome2 list (Supplementary Data 2, tab 2).

For the assignment of nucleolar RBPs, TPs were nucleolar proteins (GO:0005730) identified by OOPS RBP datasets (Supplementary Data S1, tab 3). The FPs were non-nuclear proteins without OOPS RBP annotation (Supplementary Data S1, tab 3). For each replicate, the APEX-PS-NIK3x sample was not only compared to the negative controls (e.g., omitting $H_2O_2$ or enzyme), but also compared with the APEX-PS-NLS sample. The proteins were first ranked in a descending order according to the TMT ratio and cutoff was assigned by ROC analysis as described above (Supplementary Fig. 7b). The two types of comparison were intersected for each replicate (130 C/126 C and 130 C/128 N for replicate 1; 131 N/129 C and 131 N/128 C for replicate 2; 131 C/129 C and 131 C/129 N for replicate 3). The resulting lists of the three biological replicates were intersected and the potential glycosylated proteins were removed to obtain the final nucleolar RBP list (Supplementary Data 3). For the statistical analysis of nucleolar proteins, we used a linear model (limma R package v4.1) to contrast proteins enriched from the NIK3x-targeted samples from proteins enriched in the NLS-targeted condition, taking into consideration the respective enrichment negative controls for both, effectively estimating (NIK3x − NIK3x controls) − (NLS − NLS controls). We corrected for multiple hypotheses (Benjamini–Hochberg procedure) and any proteins with an adjusted p-value of less than 0.05 were considered statistically enriched. The potential glycosylated proteins were then removed to obtain the final dataset.

For the analysis of nuclear specificity of the nuclear and nucleolar RBPs (Figs. 3g, 4f and Supplementary Figs. 5f, 8f), we collected a list of 6889 human protein (Supplementary Data 2) with nuclear annotations in the following GO terms: GO:0016604, GO:0031965, GO:0016607, GO:0005730, GO:0001650, GO:0005654, GO:0005634. The number of nuclear proteins presented in each dataset was determined. For the analysis of RNA-binding specificity of nuclear RBPs (Fig. 3f, 4g and Supplementary Fig. 5e, 8e), we collected a list of 4925 human protein (Supplementary Data 2) with RNA-binding annotations from previous studies[20,21,24,26–28,55–57] and GO annotation (GO:0003723). The number of known RBPs presented in each dataset was determined. For the sensitivity analysis of nuclear RBPome (Fig. 3h and Supplementary Fig. 5g), the gold standard nuclear RBPs was applied to determine the coverage of each method. For the sensitivity analysis of nucleolar RBPome (Supplementary Fig. 8g), a gold standard list of nucleolar RBPs was manually curated (Supplementary Data 3) according to previous literature[17] and the coverage of each dataset was determined. For comparing APEX-PS profiling with OOPS (Figs. 3i and 4h), the protein abundance of overlapped RBPs and novel RBPs identified by APEX-PS was compared according to a previous dataset[58]. The analysis of the RNA types associated with nuclear and nucleolar RBPs (Fig. 3j and Supplementary Fig. 8i) was performed as previous studies[24]. Briefly, the RBPs identified by oligodT pulldown methods[20,21,26,43,55,56] were assigned as poly(A) RNA-binding proteins. The RNA-binding types of the remaining RBPs were manually evaluated based on previous literature (Supplementary Data 2 and 3). For the analysis of RBDs, the domains of nuclear and nucleolar RBPs were obtained from Pfam (Supplementary Data 2 and 3). The classification of classical and nonclassical RBDs was based on previous studies[21,59]. The numbers of RBPs containing at least one classic RBD, only containing nonclassical RBDs or without any RBDs were determined for nuclear RBPs (Supplementary Fig. 6d). The number of nuclear RBPs containing each RBD was shown in Supplementary Fig. 6e.

**Analysis of proteomic data for OMM**. The original identified proteins are shown in Supplementary Data 4. For each replicate, the APEX-PS-OMM sample was not only compared to the negative control omitting $H_2O_2$, but also compared with the APEX-PS-NES sample. To compare APEX-PS-OMM samples to background controls, a curated list of known OMM proteins[64] was used as TPs (TP1, Sup-plementary Data 4, tab 2) and mitochondrial matrix proteins annotated by GOCC were assigned as FPs (FP1, Supplementary Data 4, tab 2). To compare APEX-PS-OMM samples to APEX-PS-NES reference controls, a curated list of known OMM proteins[64] was used as TPs (TP2, Supplementary Data 4, tab 2) and cytosolic proteins without mitochondrial annotation according to GOCC were assigned as FPs (FP2, Supplementary Data 4, tab 2). The proteins were first ranked in a descending order according to the TMT ratio and cutoff was assigned by ROC analysis as described above (Supplementary Fig. 9b). For assignment of OMM RBPs under basal condition, proteins above the cutoff of 127 C/126 C and 127 C/131 N were intersected for replicate 1 and proteins above the cutoff of 128 N/126 C and 128 N/131 N were intersected for replicate 2. For assignment of OMM RBPs under PUR treatment, proteins above the cutoff of 129 C/128 C and 129 C/131 C were intersected for replicate 1 and proteins above the cutoff of 130 N/128 C and 130 N/131 N were intersected for replicate 2. The resulting lists of the two biological replicates were intersected and the potential glycosylated proteins were removed to obtain the final OMM RBP list under basal and PUR condition, respectively (Supplementary Data 5).

For the analysis of mitochondria specificity of the OMM RBPs (Fig. 5e), a list of mitochondrial proteins were collected from MitoCarta database, GOCC terms containing mitochondrial annotations, mitochondrial matrix proteome, and IMS proteome identified by APEX profiling. The number and percentage of mitochondrial proteins in OMM RBPs under basal and PUR conditions was determined. For the analysis of OMM RBPs involved in mitochondrial-ER contact (Supplementary Fig. 9c), the number of OMM RBPs overlapped with proteins in mitochondrial-ER contact identified by split-TurboID[70] was determined. For the analysis of RNA-binding specificity of OMM RBPs (Fig. 5f), the number of known RBPs described above in OMM RBPs was determined.

**Immunofluorescence staining and fluorescence microscopy**. For fluorescence imaging experiments in Fig. 2b, HEK293T cells expressing APEX2-NLS and APEX2-NIK3x were plated and labeled as described above. Cells were fixed with 4% paraformaldehyde in PBS at room temperature for 15 min. Cells were then washed with PBS for three times and permeabilized with cold methanol at −20 °C for 5–10 min. Cells were washed again three times with PBS and blocked for 1 h with 3% BSA in DPBS ("blocking buffer") at room temperature. For APEX2-NLS imaging, cells were then incubated with anti-V5 antibody (1:1000 dilution, R960-25, ThermoFisher Scientific) in blocking buffer for 1 h at room temperature. After washing three times with DPBS, cells were incubated with DAPI/secondary anti-body (Alexa Fluor488), and neutravidin-Alexa Fluor647 (1:1000 dilution) in blocking buffer for 30 min. For APEX2-NIK3x imaging, cells were incubated with DAPI, and neutravidin-Alexa Fluor647 (1:1000 dilution) in blocking buffer for 30 min. Notably, the neutravidin-Alexa Fluor647 conjugate was synthesized as we previously reported[4]. Cells were then washed three times with DPBS and imaged. For fluorescence imaging in Fig. 4k, cells were fixed, washed and blocked as described above. Cells were incubated with anti-FBL (1:1000 dilution, ab4566, abcam) and primary antibodies for targets in blocking buffer for 1 h at room temperature. The primary antibodies for targets include anti-MIS18A (1:1000 dilution, PA5-54238, ThermoFisher Scientific) and anti-ATAD5 (1:1000 dilution, SAB4301194-100UL, Sigma–Aldrich). For fluorescence imaging in Fig. 6c, fixed cells were incubated with anti-TOMM20 (1:500 dilution, sc-17764, Santa Cruz Biotechnology) and anti-TAX1BP1 (1:500 dilution, HPA024432-100UL, Sigma–Aldrich) in blocking buffer for 1 h at room temperature. For fluorescence imaging in Supplementary Fig. 10b, HEK cells were treated with 200 μM puromycin for 30 min. Cells were fixed, washed and blocked as described above. Cells were incubated with anti-TOMM20 (1:500 dilution, sc-17764, Santa Cruz Biotechnology) and anti-SYNJ2BP (1:500 dilution, HPA000866-100UL, Sigma–Aldrich) in blocking buffer for 1 h at room temperature. After washing three times with DPBS, cells were incubated with DAPI/secondary antibody in blocking buffer for 30 min. Cells were then washed three times with DPBS and imaged.

Fluorescence confocal microscopy was performed with a Zeiss AxioObserver microscope with ×60 oil-immersion objectives, outfitted with a Yokogawa spinning disk confocal head, Cascade II:512 camera, a Quad-band notch dichroic mirror (405/488/568/647), and 405 (diode), 491 (DPSS), 561 (DPSS), and 640 nm (diode) lasers (all 50 mW). DAPI (405 laser excitation, 445/40 emission), Alexa Fluor488 (491 laser excitation, 528/38 emission), and Alexa Fluor647 (640 laser excitation, 700/75 emission) and differential interference contrast (DIC) images were acquired through a ×60 oil-immersion lens. Acquisition times ranged from 100 to 2000 ms. All images were collected and processed using SlideBook 6.0 software (Intelligent Imaging Innovations).

**Metabolic labeling of RNA–protein complexes**. For the validation of novel hits by RNA metabolic labeling (Figs. 4j and 6b), HEK293T cells were grown to ~80% confluence in 15 cm dish and treated with 1 mM 5-EU for 16 h. The cells were washed with PBS for three times, followed by irradiation with 254 nm UV light at 150 mJ/$cm^2$ (CL-1000 Ultraviolet Crosslinker, UVP). The cells were then lysed in 1 mL of 50 mM Tris-HCl (pH 7.5) buffer with sonication and subjected to cen-trifugation with 20,000 × g for 10 min to remove the debris. The lysates were reacted with 100 μM azide-PEG$_3$-biotin (Click Chemistry Tools), 500 μM $CuSO_4$, 2

mM THPTA (Sigma–Aldrich), and 5 mM sodium ascorbate (freshly prepared) for 2 h at r.t. with vortex, followed by adding 5 mM EDTA to stop the reaction. The lysates were precipitated with 8 vol of methanol at −80 °C for 1 h and washed twice with precooled methanol. The pellets were then resuspended in 1 mL RIPA lysis buffer with sonication and enriched by streptavidin beads overnight as we described above. After washing with RIPA buffer for three times, the samples were boiled in protein loading buffer with 2 mM biotin for 10 min. The samples were then analyzed by western blot with corresponding antibodies.

**RNA-immunoprecipitation sequencing (RIP-seq)**. RIP-Seq and subsequent bioinformatics analysis were done by Cloud-Seq Biotech (Shanghai, China). The SYNJ2BP RIP was performed using the GenSeqTM RIP Kit and a SYNJ2BP antibody (HPA000866-100UL, Sigma–Aldrich) along with an IgG-negative control. RNA-seq libraries were generated using the TruSeq Stranded Total RNA Library Prep Kit (Illumina) according to the manufacturer's instructions and the library quality was evaluated with BioAnalyzer 2100 system (Agilent Technologies, Inc., USA). Library sequencing was performed on an illumina Hiseq instrument with 150 bp paired-end reads. The relative enrichment of each mRNA was obtained from the fold change of gene-level FPKM (fragments per kilobase of transcript per million mapped reads) values between SYNJ2BP IP and IgG samples.

**Crosslinking immunoprecipitation (CLIP)**. To validation of SYNJ2BP mRNA targets under −PUR and +PUR conditions, HEK293T cells were treated with 0 or 200 μM puromycin for 30 min. The cells were washed with 5 mL PBS for three times, crosslinked by 254 nm UV light at 150 mJ/cm². Then the cells were lysed in 500 μL CLIP lysis buffer (50 mM Tris·Hcl, pH 7.5, 150 mM NaCl, 1% Nonidet P-40, 0.1% SDS, and EDTA-free protease inhibitor mixture). The cell lysates were incubated on ice for 10 min and cleared by centrifugation at 13000 × g for 15 min at 4 °C. 125 μL Dynabeads protein G (ThermoFisher Scientific) were washed with 500 μL lysis buffer for twice and incubated with 10 μg anti-SYNJ2BP antibody at r.t. for 45 min. The lysates were then incubated with the antibody-conjugated beads overnight at 4 °C. The beads were washed twice with 900 μL high salt wash buffer (50 mM Tris-HCl pH 7.4, 1 M NaCl, 1 mM EDTA, 1% NP-40, 0.1% SDS, and 0.5% sodium deoxycholate) and twice with wash buffer (20 mM Tris-HCl pH 7.4, 10 mM MgCl₂, 0.2% Tween-20). The beads were resuspended in 54 μL water, and then mixed with 33 μL 3× proteinase digestion buffer, 10 μL proteinase K (20 mg/mL, ThermoFisher Scientific), and 3 μL Ribolock RNase inhibitor. The 3× proteinase digestion buffer were freshly prepared as follow: 330 μL 10× PBS, pH = 7.4 (Ambion); 330 μL 20% N-laurylsarcosine sodium solution (Sigma–Aldrich); 66 μL of 0.5 M ETDA; 16.5 μL of 1 M DTT; 357.5 μL water. Proteinase digestion was performed at 42 °C for 1 h and 55 °C for 1 h with vigorous mixing and the supernatant was collected. The recovered RNAs were purified using RNA clean and concentrator −5 kit (Zymo Research) and subjected to further analysis.

**Generation of SYNJ2BP KO cells stably expressing APEX2 constructs**. The nontargeted guide and SYNJ2BP KO HEK293T cells were generated previously[64]. For preparation of lentiviruses, HEK293T cells in six-well plates were transfected at 60–70% confluency with the lentiviral vector pLX208 containing APEX2-OMM or APEX-NES (1000 ng), the lentiviral packaging plasmids dR8.91 (900 ng) and pVSV-G (100 ng), and 8 mL of Lipofectamine 2000 for 4 h. About 48 h after transfection the cell medium containing lentivirus was harvested and filtered through a 0.45 mm filter. The nontargeted guide and SYNJ2BP KO HEK293T cells were then infected at ~50% confluency, followed by selection with 250 μg/mL hygromycin in growth medium for 7 days before further analysis.

**APEX RNA labeling at OMM**. APEX labeling was performed as described above in nontargeted guide and SYNJ2BP KO HEK293T cells stably expressing APEX2-OMM or APEX2-NES. The RNA was extracted from cells using the RNeasy plus mini kit (QIAGEN) following the manufacture protocol, including adding β-mercaptoethanol to the lysis buffer. The cells were sent through the genomic DNA (gDNA) eliminator column supplied with the kit. A modification to the protocol was replacing the RW1 buffer with RWT buffer (QIAGEN) for washing. The extracted RNA was eluted into RNase-free water and RNA concentrations were determined using the Nanodrop (ThermoFischer Scientific).

To enrich biotinylated RNAs, we used 10 μL Pierce streptavidin magnetic beads (ThermoFischer Scientific) per 25 mg of RNA. The beads were washed three times in B&W buffer (5 mM Tris-HCl, pH 7.5, 0.5 mM EDTA, 1 M NaCl, 0.1% TWEEN 20), followed by two times in Solution A (0.1 M NaOH and 0.05 M NaCl), and one time in Solution B (0.1 M NaCl). The beads were then suspended in 100–150 mL 0.1 M NaCl and incubated with 100–125 mL RNA on a rotator for 2 h at 4 °C. The beads were then placed on a magnet and the supernatant discarded. Beads were washed three times in B&W buffer and resuspended in 54 mL water. A 3× proteinase digestion buffer was made (1.1 mL buffer contained 330 mL 10× PBS pH = 7.4 (Ambion), 330 μL 20% N-Laurylsarcosine sodium solution (Sigma–Aldrich), 66 mL 0.5 M EDTA, 16.5 mL 1Mdithiothreitol (DTT, ThermoFischer Scientific) and 357.5 mL water). 33 μL of this 3× proteinase buffer was added to the beads along with 10 mL Proteinase K (20 mg/mL, Ambion) and 3 mL Ribolock RNase inhibitor. The beads were then incubated at 42 °C for 1 h, followed by 55 °C for 1 h on a shaker. The RNA was then purified using the RNA clean and concentrator −5 kit (Zymo Research) and subjected to further analysis.

**RT-qPCR**. For the RT-qPCR analysis of CLIP and APEX RNA labeling experiments, the enriched RNA was first reverse transcribed following the Superscript III reverse transcriptase (ThermoFischer Scientific) protocol using random hexamers as primers. The resulting cDNA was then tested using qPCR using the primers above in 2× SYBR Green PCR Master Mix (ThermoFischer Scientific), with data generated on Lightcycler 480 (Roche). All the primer sequences used in this study are provided in Supplementary Table 1.

**Azidohomoalanine labeling**. To evaluate the impact of SYNJ2BP on protein synthesis of its clients (Figs. 6g and 7e, f), cells were cultured in methionine-free medium supplemented with 1 mM azidohomoalanine (AHA). Cells were lysed in RIPA buffer and protein concentration was normalized to 2 mg/mL. One milliliter of lysates were reacted with 100 μM biotin-PEG4-alkyne, premixed 2-(4-((bis((1-tertbutyl-1H-1,2,3-triazol-4-yl)methyl)amino)methyl)-1H-1,2,3-triazol1-yl)-acetic acid (BTTAA)-CuSO4 complex (500 μM CuSO4, BTTAA:CuSO4 with a 2:1 molar ratio) and 2.5 mM freshly prepared sodium ascorbate for 2 h at room temperature. The resulting lysates were precipitated by 8 mL methanol at −80 °C overnight and the precipitated proteins were centrifuged at 8000 × g for 5 min at 4 °C. The proteins were washed twice with 1 mL cold methanol and resuspended in 1 mL RIPA buffer with sonication. The biotinylated proteins were further captured by 200 μL streptavidin magnetic beads for 2 h. The beads were washed as described above and proteins were eluted by boiling the beads in 75 μL of 3× protein loading buffer supplemented with 20 mM DTT and 2 mM biotin. The resulting samples were analyzed by western bloting with antibodies indicated.

**Complex III and IV activity assay**. Complex III activity was assayed using a mitochondrial complex III activity assay kit (Sigma–Aldrich) and complex IV activity was determined using a complex IV human enzyme activity microplate assay kit (Abcam). HEK293T expressing nontargeted Cas9 and SYNJ2BP knockout cells were obtained from our previous study[64] were plated in 15 cm dish. Cell pellets were lysed and mitochondrion was purified according to the manufacturer's protocol in a mitochondrial isolation kit for cultured cells (Abcam). The activity was determined by following the manufacturer's protocol with a standard curve.

**Cell proliferation assays**. In order to determine the effect of SYNJ2BP on cell proliferation (Supplementary Fig. 11b), HEK293T expressing nontargeted Cas9 and SYNJ2BP knockout cells were obtained from our previous study[64]. The MTS assay was performed using the CellTiter 96 AQueous One Solution Cell Proliferation Assay kit (Promega), following the manufacturer's instructions. 1 × 10⁴ cells per well were plated in 96-well plates with 100 μL fresh medium per well. The cells were cultured for 1–3 days, and the medium was freshly changed every 24 h. 20 μL of CellTiter 96 AQueous One Solution Reagent was added into each well and incubated for 4 h.

To evaluate the impact of SYNJ2BP knockout on cell viability under CHX and PUR treatment (Fig. 7d and Supplementary Fig. 11c), 1 × 10⁴ cells per well were plated in 96-well plates with 100 μL fresh medium per well. After 24 h, the cells were treated with desired concentration (0, 25, 50, 100, 200, and 400 μM) of drugs for 12 h and then changed into normal medium for another 12 h. For glucose/galactose cell viability assay (Supplementary Fig. 11d), 2000 cells per well were plated in 96-well plates with 100 μL fresh medium per well. After 24 h, cells were washed with DPBS and the growth medium was replaced with medium containing 10% FBS, 100 units/mL penicillin, 100 mg/mL streptomycin and DMEM without glucose supplemented with 10 mM galactose or 10 mM glucose, as well as 200 μM drugs. After 24 h, the cells were changed into the mediums without drugs for 48 h and 20 μL of CellTiter 96 AQueous One Solution Reagent was added into each well and incubated for 4 h.

To evaluate the cellular recovery from heat stress (Supplementary Fig. 12g), 1 × 10⁴ cells per well were plated in 96-well plates with 100 μL fresh medium per well. After 24 h, the cells were incubated at 42 °C for 1 h and then incubated at 37 °C for 1–3 days. For the sodium arsenite stress (Supplementary Fig. 12h), cells were treated with 400 μM sodium arsenite for 1 h and then cultured in the normal medium for 1–3 days. Twenty microliters of CellTiter 96 AQueous One Solution Reagent was added into each well and incubated for 4 h. The absorbance at 490 nm was recorded using a 96-well plate reader. Each biological experiment has five technical replicates and three biological replicates were performed.

**Statistics and reproducibility**. Three biological replicates were performed for all experiments with similar results. Statistical analysis was performed on GraphPad Prism 7 (GraphPad Software). For comparison between two groups, P values were determined using two-tailed Student's t-tests, *P < 0.05; **P < 0.01; ***P < 0.001; N.S. not significant. For all boxplots (Figs. 3i, k, and 4h), P values were calculated with Wilcoxon rank sum by R (*P < 0.05; **P < 0.01; ***P < 0.001). Error bars represent means ± SD.

**Reporting summary**. Further information on research design is available in the Nature Research Reporting Summary linked to this article.

## Data availability

The proteomics data have been deposited in the MassIVE database under accession code MSV000087070. SYNJ2BP RIP-seq data have been deposited in the GEO database under accession code GSE169264. Source data are provided with this paper. Additional data beyond that provided in the Figures and Supplementary Information are available from the corresponding author upon request.

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

## Acknowledgements

We are grateful to the NIH (U24-CA210986 to S.A.C., U01-CA214125 to S.A.C., R01-DK121409 to S.A.C. and A.Y.T.), Chan Zuckerberg Biohub, and Beckman Technology Development Seed Grant for support of this work. We thank Dr. Namrata Udeshi and Dr. D. R. Mani, Broad Institute Proteomics, for their guidance on experimental design and data analysis. S. Han (Stanford University) provided images for APEX2-NLS and APEX2-NIK3x labeling.

## Author contributions

W.Q. and A.Y.T. designed the research and analyzed all the data except where noted. W.Q. performed all experiments except where noted. W.Q., A.Y.T., S.A.M., and S.A.C. designed the proteomics experiments. W.Q. prepared the proteomic samples. S.A.M. and D.K.C. processed the proteomic samples and analyzed the MS data. W.Q. and A.Y.T. wrote the paper with input from all authors.

## Competing interests

The authors declare no competing interests.
