## [Peer Review File · Nature Communications]

REVIEWER COMMENTS

Reviewer #1 (Remarks to the Author):

This manuscript by Qin et. al. describes the use of orthogonal enrichment methods in combination with spatio-temporal proximity labeling using APEX to enrich for functional subgroups of proteins. The data generated using orthogonal organic phase separation and proximity labeling (APEX-PS) is used to define a nuclear, nucleolar, and outer mitochondrial membrane (OMM) RNA binding proteome. Subsequent functional analyses of the OMM 'RBPome' using a variety of RNA profiling and electron transport chain biochemical activity assays elucidate a role for SYNJ2BP in regulating mitochondrial RNA processing independent of translation. This paper presents novel methods combining proximity labeling with functional enrichment of three protein subgroups: phosphoproteins, O-GlcNAc-proteins, and RNA binding proteins. It also uses elegant experimental design and statistical analyses to provide a thorough examination of compartment specific RBPomes, with interesting follow-up functional studies. There remain some major and minor concerns I suggest addressing. Following these, this work would be suitable for Nature Communications.

Major

Fig4

The identification of novel nucleolar RBPs is exciting, but without clear demonstration of bound mRNAs, it's possible that these are false positives. It would significantly strengthen the manuscript to identify RNA species bound to the novel RBPs Mis18A and ATAD5, similar to what was performed for SYNJ2BP.

Fig5

The rationale for fewer proteins binding mRNA in +PUR samples is unclear. It could be argued that, given the disassembly of polysomes and the lack of translation, there could be increased mRNA binding in this scenario. Both scenarios are equally valid and should be considered in the manuscript. Figures 5C,5D elucidate that a subset of RBPs do not bind RNA in +PUR conditions. A bioinformatic follow-up study (e.g., assessing GO terms, specifically molecular function) may shed light on whether there are properties common amongst this group which differentiates them from RNA binding proteins in -PUR/+PUR conditions and why they are translation dependent RNA binding proteins.

Fig6

In looking for direct clients, the authors state that >100 mRNAs were enriched for SYNJ2BP. They then select 11 of these, but it is unclear why these were chosen. In supplemental table 6, these 11 rank as 1, 4, 16, 17, 20, 24, 31, 34, 43, 45, and 104 in terms of fold change. Why do these 11 in particular warrant further analysis? Relatedly, the authors go on to show that 5 of the 11 selected mRNAs were significantly reduced at the OMM when SYNJ2BP was absent. What about the others 93?

Fig7

The authors show Complex III and Complex IV deficiency in SYNJ2BP cells recovering from PUR treatment and correlate this significant reduction in activity with the quantitative reduction in a single member of either CIII or CIV. It would be beneficial to see if complex assembly is affected in this condition using native conditions. The lack of UQCR11 and PET117 should disrupt CIII and CIV assembly respectively and would provide more support for the mechanistic conclusions drawn from this work.

Minor

FigS3A

The mitochondrial and ER labelling appears reversed.

Line372

Reference has not been correctly annotated.

Line439

SYNJ2BP has a typo (SYJN2JBP).

Reviewer #2 (Remarks to the Author):

In this manuscript, Qin et al combine APEX for protein proximity labeling with enrichment of functional protein classes to characterize the presence of specific protein subclasses within distinct organelles. Specifically, they biotinylated proteins in the nucleus, in nucleoli, or the outer mitochondrial membrane (OMM), and enrich phosphorylated, glycosylated, or RNA-binding proteins showing the presence of proteins that are known to be modified (or to bind RNA) at the targeted cellular locations. In addition they identify various novel RNA-binding proteins (RBPs) in the OMM, which were validated by independent methods. The authors demonstrate that SYNJ2BP is a novel RBP that directs specific mRNAs at the OMM for their local translation during stress recovery.

This manuscript addresses an important question bridging the gap between the ability to determine protein localization and to chart protein interactions/modifications. Several methods have been developed in recent years to determine protein composition of organelles (e.g. using proximity labeling techniques such as APEX), and to globally profile protein modifications and interactions. Yet, a recurrent question in any of these approaches is to understand how localization/modification relates to protein function. The current manuscript does not immediately solve this question, but brings this a step closer by investigating the presence of functional protein classes in distinct organelles. This arguably helps to prioritize candidates in subsequent functional experiments. Technically, the authors approach this by combining APEX tools that had been previously developed to biotinylate proteins in different organelles (nucleus, nucleolus, OMM) with the enrichment of various protein classes (phosphorylated, glycosylated, RNA-binding proteins). This results in a somewhat heterogeneous and non-focused study, showing proof-of-principle for several scenarios (phosphorylated proteins in the nucleus, RBPs in the nucleolus), not going beyond showing expected behavior of positive controls. In fact these parts of the manuscript stop before they get really interesting. The exception is the last section in the manuscript where the authors go into significant depth to show how the novel RBP SYNJ2BP mediates local translation of distinct mRNAs at the OMM. Technically, the shown experiments were performed at an advanced level, however I have some concerns about how the authors treat their data to classify enriched proteins.

A number of general and specific comments can be made:

1. Fig 1a and associated data: APEX is used here to biotinylate (all) nuclear proteins. How is this better, or what are the advantages, compared to performing a nuclear prep? E.g. with regard to sensitivity, specificity, number of cells needed?
2. Their functional PL approach relies on the combination of several procedures, each of which are known to be inefficient, making me wonder how many cells are typically needed for these experiments (methods section mentions the requirement of several mg of protein). For instance, phosphorylation is rare, biotinylation incomplete, IMAC inefficient, and streptavidin enrichment incomplete. When multiplying these effects, how does this affect the amount of input material to obtain informative data in the end? This should be stated more explicitly to get an understanding of the required scale of the experiments/methods.
3. Fig 1e: Why was this done for 1 target only (ERK), and why was it done by western blot only? It would have been interesting to analyse the enriched sample by MS to globally profile the nuclear phospho-proteome. This should be possible for these labs with relatively little additional effort, and

I'm almost certain that the authors have tried this. Why was it unsuccessful? Currently the data only show a snapshot showing a few positive controls with no new information. It is somewhat disappointing that this PTM story ends here before it even gets started?

4. The authors apply formaldehyde (FA) as well as UV light for crosslinking, however they use some odd reasoning why they settle on FA for the majority of analyses: they say that FA is 'specific enough' (whatever that means) to study protein-RNA interactions since it was used to study protein-DNA (!) interactions in the past (line 190). In addition, authors talk around the presumed equality of FA and UV-induced capture of RNA-binding proteins: FA is more efficient than UV (line 195), with the same specificity (line 196), however UV is still more 'stringent' (Line 202)? These arguments do not add up, so it is unclear what this means and how this leads to the preferred use of FA (apart from the fact that conveniently this yields more protein).

5. Suppl fig 4a: In each of the shown plots, data do not seem to be normally distributed ('hanging' below the diagonal). Were data treated/normalized in any particular way before plotting? Instead or in addition to showing this as scatter plots, it will be insightful to see the same data in a histogram. If this indeed indicates non-normal distribution of data, authors should explain how this happened and how it affects data interpretation.

6. Suppl fig 4b: ROC curves look rather poor, almost a diagonal suggesting that no enrichment occurred. Apparently many non-nuclear were co-enriched as false positives. Alternatively, proteins that were deemed non-nuclear may still localize to the nucleus. Therefore, applying a cut-off ratio (suppl fig 4c) is both biased and a random choice, and it risks the exclusion of true positives.

7. Line 262: in a way it is unfortunate that random cutoffs combined with using prior knowledge gives better results (or at least higher numbers) compared to applying unbiased statistics based on robust experimental design. It would have been insightful to investigate the source of this discrepancy and find an appropriate solution. As a first step, some indication may be gained by plotting the 'RBPome1' data on top of the volcano plot in Suppl Fig 5a to see if the majority of proteins tends to be enriched (without reaching significance).

8. Related to the previous point, having low ratios (i.e. poor enrichment, Suppl fig 4c) may also indicate that quantification was performed at low S/N levels, i.e. even in biotinylated samples the amount of recovered protein was low. It is not surprising if this occurs, taking into account the sparsity of biotinylation, and relative low efficiency of protein retrieval in a dual enrichment procedure (also see point 2 above). Instead of removing proteins below a (random) ratio, it would have made more sense to exclude proteins falling below a S/N level that prohibit reliable quantification.

9. Line 236: There is a suspiciously large amount of manual data filtering, removing proteins below a certain (non-justified) ratio (suppl fig 4c) and glycoproteins. Why can the latter not bind RNA?

10. Suppl fig 5a: How can it be explained that not only proteins were enriched in the sample over background (as expected), but also in background over sample (left side of the figure) for a similar or even larger number of proteins? This indicates that proteins are selectively enriched without biotinylation and/or crosslinking (depending on what is meant by 'background'). This requires an explanation in the manuscript.

11. Line 141: Fig 1d is a cartoon (not data), hence the wording in this sentence should be changed (nuclear translocation was not 'shown' and cannot be referred to as a 'finding').

Reviewer #3 (Remarks to the Author):

In this manuscript by Qin et al., the authors extend APEX2-mediated proximity labeling (PL) to mapping subcellular functional proteome, with a particular focus on RNA-binding proteomes (RBPomes). In the first part, the authors combine APEX labeling with IMAC- and WGA-based affinity purification workflow to study nuclear phosphoproteome and glycoproteome (O-GlcNAcylation), respectively. The author demonstrate that the fast kinetics of APEX labeling could resolve the dynamic of protein phosphorylation and nuclear import. In the second part, the authors focus on subcellular RBPome profiling, in which APEX labeling is integrated with OOPS-mediated RBP extraction (APEX-PS). The authors employ APEX-PS to map nuclear, nucleolar, cytosolic, and outer-mitochondrial membrane

(OMM) RBPs, and identify SYNJ2BP as a novel RNA anchoring protein on the OMM. They further verify that SYNJ2BP facilitates functional recovery of mitochondria after stress. Overall, APEX-mediated PL coupled with functional enrichment represents a simple yet powerful approach for identifying subcellular proteins with versatile functions. The manuscript is well written and the data are clearly presented. I recommend publication in Nature Communication pending the following minor revisions:

1. In its current form, the first part - "Compartment-specific enrichment of phosphorylated and O-GlcNAcylated proteins" - seem quite disconnected to the remaining text of the article, which focused heavily on the RBPome profiling and follow-up RBP studies. The authors may consider including a cartoon illustration that summarizes these diverse applications at the beginning (e.g. Fig. 1), so that readers could better appreciate the broad applicability of functional proximity labeling.
2. Also in this part, the authors characterize the specificity of tandem enrichment with western blotting of proteins with known posttranslational modifications and subcellular localization, rather than mass spectrometry. This part could be strengthened if proteomic data are shown, which may reveal novel nuclear phosphor- and glyco-proteins. However, since this part is not directly linked with the RBP story, I consider this as optional but not required.
3. To what extent does APEX labeling, especially 1 min H₂O₂ treatment, affect protein phosphorylation and O-GlcNAcylation? The authors could address the question by quantifying global protein modification level pre- and post-APEX labeling reaction.
4. How would the sequence of APEX labeling and FA treatment affect the outcome of RBP profiling? In the previous APEX-RIP paper, the authors compared the APEX-first and FA-first workflow. A similar comparison could be performed here. It is also important for the readers to know that the sequence is important for high data quality.
5. Related to lines 77-78: "non-membrane enclosed regions that are impossible to purify by biochemical fractionation – including the nucleolus and outer mitochondrial membrane (OMM)", the statement is not entirely accurate since nucleolus has been previously purified biochemically (PMID: 11790298)
6. Related to lines 231-232: "We applied the TMT ratio cutoff that maximized retention of true positives and minimized retention of false positives for each dataset". This statement is not entirely accurate, as in theory, the maximization of TP retention and the minimization of FP retention are two contradictory goals. In a ROC analysis, one typically wants to maximize the difference between TP and FP rates.
7. Related to lines 240-241: "The second approach we used to filter the MS data consists of two-sample T-tests to compare APEX-NLS replicates to background controls" and Supplementary Fig. 5a. Please clarify "background controls".
8. Related to lines 435-438: "To determine if these 11 mRNAs are localized to the OMM solely through the action of SYNJ2BP or if other binding interactions also play a role, we used OMM-localized APEX to directly biotinylate the RNAs at the OMM in both wild-type and SYNJ2BP knock-out HEK293T cells", the effect of SYNJ2BP KO can demonstrate that SYNJ2BP participates in the OMM targeting of these mRNAs, but cannot tell whether SYNJ2BP alone or together with other interacting proteins anchors these mRNAs at the OMM. For example, SYNJ2BP may function as a protein anchor for other RBPs. This point could be discussed briefly in the main text.
9. In Supplementary Fig. 5f, "Not annotaed" should be "Not annotated".
10. Line 431: "SYNJBP" should be "SYNJ2BP".
11. Lines 313-314: "where proteins are extensively and dynamically interact with RNAs" should be "where proteins are extensively and dynamically interacting with RNAs".

Point-by-point responses to reviewer comments

Below are our point-to-point responses to the questions and concerns raised by the reviewers (colored in blue). The newly added or revised texts have also been colored in yellow in the manuscript files to facilitate the second round of review.

Reviewer #1:

“This manuscript by Qin et. al. describes the use of orthogonal enrichment methods in combination with spatio-temporal proximity labeling using APEX to enrich for functional subgroups of proteins. The data generated using orthogonal organic phase separation and proximity labeling (APEX-PS) is used to define a nuclear, nucleolar, and outer mitochondrial membrane (OMM) RNA binding proteome. Subsequent functional analyses of the OMM ‘RBPome’ using a variety of RNA profiling and electron transport chain biochemical activity assays elucidate a role for SYNJ2BP in regulating mitochondrial RNA processing independent of translation. This paper presents novel methods combining proximity labeling with functional enrichment of three protein subgroups: phosphoproteins, O-GlcNAc-proteins, and RNA binding proteins. It also uses elegant experimental design and statistical analyses to provide a thorough examination of compartment specific RBPomes, with interesting follow-up functional studies. There remain some major and minor concerns I suggest addressing. Following these, this work would be suitable for Nature Communications.”

Thank you for appreciating our work.

“Major/Fig4: The identification of novel nucleolar RBPs is exciting, but without clear demonstration of bound mRNAs, it’s possible that these are false positives. It would significantly strengthen the manuscript to identify RNA species bound to the novel RBPs Mis18A and ATAD5, similar to what was performed for SYNJ2BP.”

We thank the reviewer for pointing this out. We have already validated the two novel nucleolar RBPs Mis18A and ATAD5 using an orthogonal, established RNA binding assay, which is 5-EU labeling followed by UV crosslinking in **Figure 4j**. We furthermore performed imaging of endogenous Mis18A and ATAD5 to verify their nucleolar localization in **Figure 4k** (both proteins are not annotated as nucleolar proteins in GOCC). This meets the standard in the field, which is to validate any hits identified by a novel method using other/orthogonal/more traditional methods. Based on our data, we have confidence that the novel nucleolar RBPs we validated do indeed bind to RNA and are localized to the human nucleolus; digging into their specific RNA clients is beyond the scope of this study given that we have focused our deep dive in the second half of our paper on mitochondrial membrane RBPs and SYNJ2BP in particular.

“Fig5: The rationale for fewer proteins binding mRNA in +PUR samples is unclear. It could be argued that, given the disassembly of polysomes and the lack of translation, there could be increased mRNA binding in this scenario. Both scenarios are equally valid and should be considered in the manuscript. Figures 5C,5D elucidate that a subset of RBPs do not bind RNA

in +PUR conditions. A bioinformatic follow-up study (e.g., assessing GO terms, specifically molecular function) may shed light on whether there are properties common amongst this group which differentiates them from RNA binding proteins in -PUR/+PUR conditions and why they are translation dependent RNA binding proteins.”

The reason we expected fewer RBPs in the +PUR condition was that our previous APEX-seq study¹ showed far fewer mRNAs at the OMM following +PUR treatment. However, it is certainly possible that some OMM-localized RBPs bind mRNA more following PUR treatment than before. In fact, vimentin (VIM, an intermediate filament protein known to associate with mitochondria) was one such hit (**Figure R1** below, **Supplementary Figure 9d**).

We did perform analyses to check for any features common to the RBPs we enriched in the -PUR condition but not +PUR conditions (translation-dependent RBPs). For example, do these RBPs have domains that interact with both mRNAs and ribosomes to assist with local translation at the OMM? Are they weaker-binding or tighter-binding RBPs? Unfortunately, due to the small number of hits, we did not detect any clear pattern.

Figure R1. Increased RNA binding by VIM at OMM upon PUR treatment. The change in RNA binding extent for other ribosome-independent OMM-localized RBPs is also shown. This figure is shown as **Supplementary Figure 9d** in the revised manuscript.

“Fig6: In looking for direct clients, the authors state that >100 mRNAs were enriched for SYNJ2BP. They then select 11 of these, but it is unclear why these were chosen. In supplemental table 6, these 11 rank as 1, 4, 16, 17, 20, 24, 31, 34, 43, 45, and 104 in terms of fold change. Why do these 11 in particular warrant further analysis? Relatedly, the authors go on to show that 5 of the 11 selected mRNAs were significantly reduced at the OMM when SYNJ2BP was absent. What about the others 93?”

We have clarified in the revised manuscript that we selected these 11 RIP-seq hits for follow-up analysis because they encode essential mitochondrial proteins. Because we have a biological interest in local translation at the mitochondrial membrane and RBPs which may be involved in this process, we focused on mRNA transcripts encoding important

mitochondrial proteins. Five of these 11 selected hits (UQCR11, PET117, RAB5IF, ATP5E and ATP5O) are directly linked with OXPHOS and three (TRIT1, PDPR and ACS1) are involved in mitochondrial metabolism. Moreover, we deliberately selected mRNAs across a range of enrichment ratios to better assess the validity of our RIP-seq list.

We only performed follow-up on these 11 hits (by qPCR), not the 93 other mRNAs in the RIP-seq dataset. As shown in **Figure 6f** and **Supplementary Figure 10d**, 5 of the 11 hits (with higher RIP-seq enrichment ratios) were reduced at the OMM when SYNJ2BP was absent, whereas the other 6 mRNAs with lower RIP-seq enrichment scores were not affected by SYNJ2BP KO. This suggests that the most highly enriched mRNAs in our RIP-seq dataset are more likely to be regulated by SYNJ2BP.

“Fig7: The authors show Complex III and Complex IV deficiency in SYNJ2BP cells recovering from PUR treatment and correlate this significant reduction in activity with the quantitative reduction in a single member of either CIII or CIV. It would be beneficial to see if complex assembly is affected in this condition using native conditions. The lack of UQCR11 and PET117 should disrupt CIII and CIV assembly respectively and would provide more support for the mechanistic conclusions drawn from this work.”

We have performed new experiments following the reviewer’s suggestion. **Figure R2** below (**Figure 7c** in the revised manuscript) shows that assembly of both Complex III and Complex IV are significantly disrupted in the absence of SYNJ2BP during the recovery process from PUR stress. By comparison, the assembly of Complexes II and V are not affected. Interestingly, the assembly of Complex I was also decreased to some extent, perhaps due to the contribution of other SYNJ2BP clients such as RAB5IF (Respirasome Complex Assembly Factor 1). Our new data further support the conclusion that SYNJ2BP can promote the recovery of OXPHOS activity and assembly by promoting the translation of its mRNA clients.

Figure R2. Evaluation of Complex I-V assembly 12 hours following treatment of HEK cells with protein translation inhibitor PUR. Whole cell lysates were probed with total OXPHOS antibody cocktail against subunits that are labile when their corresponding complexes are not assembled².

³. Quantification shown from three biological replicates. This figure is shown as **Figure 7c** in the revised manuscript.

“Minor/FigS3A: The mitochondrial and ER labelling appears reversed.

Line372: Reference has not been correctly annotated.

Line439: SYNJ2BP has a typo (SYJN2JBP).”

Thank you for pointing these out. We have corrected them.

Reviewer #2:

“In this manuscript, Qin et al combine APEX for protein proximity labeling with enrichment of functional protein classes to characterize the presence of specific protein subclasses within distinct organelles. Specifically, they biotinylated proteins in the nucleus, in nucleoli, or the outer mitochondrial membrane (OMM), and enrich phosphorylated, glycosylated, or RNA-binding proteins showing the presence of proteins that are known to be modified (or to bind RNA) at the targeted cellular locations. In addition they identify various novel RNA-binding proteins (RBPs) in the OMM, which were validated by independent methods. The authors demonstrate that SYNJ2BP is a novel RBP that directs specific mRNAs at the OMM for their local translation during stress recovery.

This manuscript addresses an important question bridging the gap between the ability to determine protein localization and to chart protein interactions/modifications. Several methods have been developed in recent years to determine protein composition of organelles (e.g. using proximity labeling techniques such as APEX), and to globally profile protein modifications and interactions. Yet, a recurrent question in any of these approaches is to understand how localization/modification relates to protein function. The current manuscript does not immediately solve this question, but brings this a step closer by investigating the presence of functional protein classes in distinct organelles. This arguably helps to prioritize candidates in subsequent functional experiments. Technically, the authors approach this by combining APEX tools that had been previously developed to biotinylate proteins in different organelles (nucleus, nucleolus, OMM) with the enrichment of various protein classes (phosphorylated, glycosylated, RNA-binding proteins). This results in a somewhat heterogeneous and non-focused study, showing proof-of-principle for several scenarios (phosphorylated proteins in the nucleus, RBPs in the nucleolus), not going beyond showing expected behavior of positive controls. In fact these parts of the manuscript stop before they get really interesting. The exception is the last section in the manuscript where the authors go into significant depth to show how the novel RBP SYNJ2BP mediates local translation of distinct mRNAs at the OMM. Technically, the shown experiments were performed at an advanced level, however I have some concerns about how the authors treat their data to classify enriched proteins.”

We thank the reviewer for the positive evaluation of our manuscript.

“A number of general and specific comments can be made:

1. Fig 1a and associated data: APEX is used here to biotinylate (all) nuclear proteins. How is

this better, or what are the advantages, compared to performing a nuclear prep? E.g. with regard to sensitivity, specificity, number of cells needed?"

Following the reviewer's suggestion, we have generated a new figure in which nuclear APEX labeling is compared side by side with nuclear fractionation (Figure R3 below, and Supplementary Figure 2g in the revised manuscript). In each case, we started with $\sim 5 \times 10^7$ cells. For fractionation-PS, the FA-crosslinked cells were subjected to nuclear fractionation based on established protocols⁴ followed by the standard OOPS protocol to enrich RBPs. The specificity and sensitivity of both methods were evaluated by blotting against the protein markers we previously used (in Figure 2d). Both methods are specific, enriching the nuclear RBPs SRSF1 and hnRNPC, but not the mitochondrial RBP GRSF1 nor the nuclear non-RBP ETS2. In terms of sensitivity, we found that the enrichment of SRSF1 and hnRNPC by APEX-PS-NLS was somewhat higher than enrichment of the same proteins by nuclear fractionation-PS. This may be because in the latter case, PS is performed after nuclear fractionation and mRNAs may become degraded to some extent during fractionation. By contrast, for all versions of functional proximity labeling shown in this paper, functional enrichment of protein subclasses was performed immediately after cell lysis to maximize recovery.

We wish to emphasize that even though nuclear RBPs or PTMs can be analyzed via nuclear fractionation, the main advantage/utility of functional proximity labeling will be when probing subcellular compartments that are difficult or impossible to purify by fractionation, such as the nucleolus and OMM.

Figure R3. Comparison of APEX-PS-NLS and nuclear fractionation followed by OOPS (nuclear fractionation-PS). Both methods start with the same number of starting cells. The enriched materials were blotted against nuclear RBPs (SRSF1 and hnRNPC), a mitochondrial RBP (GRSF1) and a nuclear non-RBP (ETS2). This data is shown as Supplementary Figure 2g in the revised manuscript.

"2. Their functional PL approach relies on the combination of several procedures, each of which are known to be inefficient, making me wonder how many cells are typically needed for these experiments (methods section mentions the requirement of several mg of protein). For instance, phosphorylation is rare, biotinylation incomplete, IMAC inefficient, and streptavidin enrichment incomplete. When multiplying these effects, how does this affect the

amount of input material to obtain informative data in the end? This should be stated more explicitly to get an understanding of the required scale of the experiments/methods.”

We have edited the method section to state explicitly the scale of our functional proximity labeling experiments. For APEX-PS, APEX-IMAC and APEX-WGA, we begin with 5×10^7 cells from a 15-cm dish at ~90% confluency. This is a reasonable scale for most experimentalists and provides sufficient final enriched material for both western blot and mass spec proteomics analysis.

“3. Fig 1e: Why was this done for 1 target only (ERK), and why was it done by western blot only? It would have been interesting to analyse the enriched sample by MS to globally profile the nuclear phospho-proteome. This should be possible for these labs with relatively little additional effort, and I'm almost certain that the authors have tried this. Why was it unsuccessful? Currently the data only show a snapshot showing a few positive controls with no new information. It is somewhat disappointing that this PTM story ends here before it even gets started?”

We have not attempted proteomics experiments using APEX-IMAC or APEX-WGA protocols and are not withholding any unsuccessful proteomics data. Based on the pilot experiments shown in **Figure 1**, we believe it is highly likely that APEX-IMAC or APEX-WGA proteomics would work well and is worth pursuing in future studies.

The reason why we did not perform proteomic experiments with APEX-IMAC and APEX-WGA in this study is that we do not believe it is useful to obtain and publish proteomic datasets without follow up validation to show that the proteomics data are of high quality. Since this is very labor intensive, we opted to do this for just one of the functional proximity labeling procedures introduced in this work, APEX-PS for discovery of subcompartment-specific RBPs. For the two other flavors of functional PL shown in this work, APEX-IMAC and APEX-WGA, we evaluated the methods using WB detection of true positives and true negatives in **Figure 1c-d** and then turned our full attention to APEX-PS and RBP discovery for the remainder of the manuscript, performing proteomics, follow up validations, and biological investigation of a single mitochondrial RBP hit (SYNJ2BP).

“4. The authors apply formaldehyde (FA) as well as UV light for crosslinking, however they use some odd reasoning why they settle on FA for the majority of analyses: they say that FA is 'specific enough' (whatever that means) to study protein-RNA interactions since it was used to study protein-DNA (!) interactions in the past (line 190). In addition, authors talk around the presumed equality of FA and UV-induced capture of RNA-binding proteins: FA is more efficient than UV (line 195), with the same specificity (line 196), however UV is still more 'stringent' (Line 202)? These arguments do not add up, so it is unclear what this means and how this leads to the preferred use of FA (apart from the fact that conveniently this yields more protein).”

We apologize for our confusing writing and have edited the manuscript text to make it clearer. While there is general agreement in the literature that FA crosslinking is not as specific as UV crosslinking, previous studies have shown that FA can generate quite specific RBP datasets⁵⁻⁸, and the specific protein markers we selected for our comparisons in **Supplementary Figure 2e** did not show a specificity difference between FA and UV. Because we found that FA is much higher yielding than UV we opted for FA instead of UV. In other words, we traded much higher sensitivity for a small drop in specificity.

“5. Suppl fig 4a: In each of the shown plots, data do not seem to be normally distributed ('hanging' below the diagonal). Were data treated/normalized in any particular way before plotting? Instead or in addition to showing this as scatter plots, it will be insightful to see the same data in a histogram. If this indeed indicates non-normal distribution of data, authors should explain how this happened and how it affects data interpretation.”

The data were not treated or normalized in any way. The majority of proteins were clustered close to the diagonal and only a few points were dispersed below the diagonal. We have replotted the data as histograms, per the reviewer’s suggestion, and find that they are normally distributed (**Figure R4**, and new **Supplementary Figure 4b**).

Figure R4. Histogram of relative ratios between biological replicates, including 128C/128N, 129N/128N and 129N/128C. This figure is shown as **Supplementary Figure 4b** in the revised manuscript.

“6. Suppl fig 4b: ROC curves look rather poor, almost a diagonal suggesting that no enrichment occurred. Apparently many non-nuclear were co-enriched as false positives. Alternatively, proteins that were deemed non-nuclear may still localize to the nucleus. Therefore, applying a cut-off ratio (suppl fig 4c) is both biased and a random choice, and it risks the exclusion of true positives.”

We agree that the ROC curves do not look very good, and we believe it is because it is difficult to generate truly accurate TP and FP protein lists of nuclear and non-nuclear proteins respectively. Many proteins annotated as “nuclear” in GOCC actually have protein subpopulations in the cytosol, and vice versa.

We repeated our ROC analysis using more stringent TP and FP lists. For TP, we used highly validated “gold standard” nuclear RBPs such as splicing factors and ribosomal proteins (full list of 155 TP proteins shown in Supplementary Table 1). For FP, we used mitochondrial

matrix proteins that are very unlikely to also be present in the nucleus. As shown in **Figure R5** and new **Supplementary Figure 4c**, the resulting ROC curves look better than before, and the difference between TPR and FPR improves. With the new cut-off ratios, we generated a final list of 791 nuclear RBPs, which are largely overlapping with the 764 nuclear RBPs previously obtained (714 proteins in common). We have revised the manuscript to use this new nuclear RBP dataset, which also displays excellent sensitivity and specificity (**Figure 3f-h**).

Figure R5. ROC curves generated using a more stringent set of TPs and FPs. A list of highly validated gold standard nuclear RBPs was used as TPs and a list of mitochondrial matrix proteins was used as FPs. This data is shown as **Supplementary Figure 4c** in the revised manuscript.

“7. Line 262: in a way it is unfortunate that random cutoffs combined with using prior knowledge gives better results (or at least higher numbers) compared to applying unbiased statistics based on robust experimental design. It would have been insightful to investigate the source of this discrepancy and find an appropriate solution. As a first step, some indication may be gained by plotting the 'RBPome1' data on top of the volcano plot in Suppl Fig 5a to see if the majority of proteins tends to be enriched (without reaching significance).”

We disagree that our ROC-based cutoff method is “random”. There is a clear inflection point in the ranked data above which TPs are enriched and below which FPs are enriched. Following the reviewer’s suggestion, we plotted RBPome1 data on top of the volcano plot in **Supplementary Figure 5a**. **Figure R6** shows that RBPome1 proteins (red) are overwhelmingly right-shifted in this plot.

Figure R6. The distribution of RBPome1 proteins (red) in the volcano plot from **Supplementary Figure 5a**.

“8. Related to the previous point, having low ratios (i.e. poor enrichment, Suppl fig 4c) may also indicate that quantification was performed at low S/N levels, i.e. even in biotinylated samples the amount of recovered protein was low. It is not surprising if this occurs, taking into account the sparsity of biotinylation, and relative low efficiency of protein retrieval in a dual enrichment procedure (also see point 2 above). Instead of removing proteins below a (random) ratio, it would have made more sense to exclude proteins falling below a S/N level that prohibit reliable quantification.”

Our data are not filtered by an “S/N” metric, but we take many precautions to ensure that only high quality data makes it into final analyses and interpretation: From an analytical chemistry standpoint, we only run at high caliber chromatographic separations and recently calibrated mass accuracy. We have metrics we constantly observe to maintain high standards. We fractionate the sample to concentrate analytes, and to reduce the TMT interference. For peptide and protein identifications and quantitation, we roll up peptide spectral matches (PSMs) in a way that minimizes outlier PSMs. PSMs with poor quality metrics, from which low intensity ions is often the cause, are discarded. We only include proteins identified with at least two peptides (two independent measurements) for subsequent analyses. For statistical testing we include moderation which models in, and accounts for, the variance within each sample. We also always correct for multiple hypothesis testing, adding another step of rigor.

Also, we note that despite the reviewer’s fears that we recovered a miniscule amount of protein after double enrichment, our silver-stained protein gels show substantial protein recovery: please compare the sample lanes (lane #8 in **Supplementary Figure 2a**) compared to the negative control lanes (lanes #6,7) in which APEX or H₂O₂ was omitted.

“9. Line 236: There is a suspiciously large amount of manual data filtering, removing proteins

below a certain (non-justified) ratio (suppl fig 4c) and glycoproteins. Why can the latter not bind RNA?”

We respectfully disagree and believe that our ROC analysis based on known TPs and FPs to generate cut-offs is justified, unbiased, and also well-established in the literature⁹⁻¹³. Regarding manual removal of glycoproteins, while it is formally possible that some glycoproteins may also be RBPs, the vast majority are not, and yet glycoproteins migrate to the interphase in organic-aqueous phase separation and contaminate the RBPs there. The published OOPS protocol we are following¹⁴ also manually removed glycoproteins from their final datasets. For each APEX-PS dataset (Supplementary Tables 2, 3, 5), we have included a tab showing the glycoproteins we removed so that the reader can inspect these lists if they are interested in taking a deeper look.

“10. Suppl fig 5a: How can it be explained that not only proteins were enriched in the sample over background (as expected), but also in background over sample (left side of the figure) for a similar or even larger number of proteins? This indicates that proteins are selectively enriched without biotinylation and/or crosslinking (depending on what is meant by 'background'). This requires an explanation in the manuscript.”

Statistical enrichment in the negative direction is common for enrichment experiments where the readout is LCMS¹⁵. Abundant proteins are usually enriched in the negative direction because they bind the beads, or survive the enrichment protocol. There will be a bias for high abundance proteins, and hydrophobic proteins that tend to bind nonspecifically to beads. When there is nothing to pulldown, like in a negative control, there will be more background binders. They are still detected and quantified, but they are depleted from the experimental condition, aka, enriched in the background.

We can observe the effects of quantifying non-specific proteins by the data structure. There are many proteins enriched in the negative direction that have large average fold-changes but do not reach statistical significance. This indicates their variance was too high to pass the cutoff. In contrast, the enrichment side (right) increases more monotonically—as the fold change increases so does the statistical significance. This indicates less variance and thus more specificity to the experimental enrichment.

“11. Line 141: Fig 1d is a cartoon (not data), hence the wording in this sentence should be changed (nuclear translocation was not ‘shown’ and cannot be referred to as a 'finding').”

We have edited this accordingly.

Reviewer #3:

“In this manuscript by Qin et al., the authors extend APEX2-mediated proximity labeling (PL) to mapping subcellular functional proteome, with a particular focus on RNA-binding proteomes (RBPomes). In the first part, the authors combine APEX labeling with IMAC- and WGA-based affinity purification workflow to study nuclear phosphoproteome and

glycoproteome (O-GlcNAcylation), respectively. The author demonstrate that the fast kinetics of APEX labeling could resolve the dynamic of protein phosphorylation and nuclear import. In the second part, the authors focus on subcellular RBPome profiling, in which APEX labeling is integrated with OOPS-mediated RBP extraction (APEX-PS). The authors employ APEX-PS to map nuclear, nucleolar, cytosolic, and outer-mitochondrial membrane (OMM) RBPs, and identify SYNJ2BP as a novel RNA anchoring protein on the OMM. They further verify that SYNJ2BP facilitates functional recovery of mitochondria after stress. Overall, APEX-mediated PL coupled with functional enrichment represents a simple yet powerful approach for identifying subcellular proteins with versatile functions. The manuscript is well written and the data are clearly presented. I recommend publication in Nature Communication pending the following minor revisions:”

Thank you for appreciating our work.

1. In its current form, the first part - “Compartment-specific enrichment of phosphorylated and O-GlcNAcylated proteins” - seem quite disconnected to the remaining text of the article, which focused heavily on the RBPome profiling and follow-up RBP studies. The authors may consider including a cartoon illustration that summarizes these diverse applications at the beginning (e.g. Fig. 1), so that readers could better appreciate the broad applicability of functional proximity labeling.”

Thank you for the suggestion. We have included a new schematic (Figure R7 and Figure 1a) that summarizes functional proximity labeling in multiple forms.

Rebuttal figure 7. The workflow of functional PL that combines APEX-catalyzed PL with functional protein enrichment (e.g. PS, IMAC or WGA) and streptavidin bead capture to enrich subcellular protein subclasses. This figure is shown as Figure 1a in the revised manuscript.

“2. Also in this part, the authors characterize the specificity of tandem enrichment with western blotting of proteins with known posttranslational modifications and subcellular localization, rather than mass spectrometry. This part could be strengthened if proteomic data are shown, which may reveal novel nuclear phosphor- and glyco-proteins. However, since this part is not directly linked with the RBP story, I consider this as optional but not

required.”

Indeed we have focused our efforts on fleshing out the RBP part of our manuscript, performing proteomics under two different cellular conditions and then investigating in-depth our newly discovered mitochondrial RBP SYNJ2BP.

“3. To what extent does APEX labeling, especially 1 min H₂O₂ treatment, affect protein phosphorylation and O-GlcNAcylation? The authors could address the question by quantifying global protein modification level pre- and post-APEX labeling reaction.”

We performed a new experiment to evaluate this question (**Figure R8** below, **Supplementary Figure 1b-c** in the revised manuscript) and found that both phosphorylation and O-GlcNAcylation (on a global scale) are not noticeably affected by APEX + 1 min H₂O₂ treatment.

Figure R8. The impact of 1 min H₂O₂ treatment on the global level of phosphorylation and O-GlcNAcylation. **a**, The impact of 1 min H₂O₂ treatment on global phosphorylation level detected by antibodies against phosphorylation on serine/threonine and tyrosine. **b**, The impact of 1 min H₂O₂ treatment on global O-GlcNAcylation level detected by an O-GlcNAc antibody (RL2). This data is shown as **Supplementary Figure 1b-c** in the revised manuscript.

“4. How would the sequence of APEX labeling and FA treatment affect the outcome of RBP profiling? In the previous APEX-RIP paper, the authors compared the APEX-first and FA-first workflow. A similar comparison could be performed here. It is also important for the readers to know that the sequence is important for high data quality.”

Following the reviewer’s suggestion, we have performed a side-by-side comparison of the biotinylation/APEX-first procedure versus the FA-first procedure. **Figure R9** below and new **Supplementary Figure 2f** show that while the sensitivity of both workflows are comparable (the nuclear RBPs SRSF1 and hnRNPC are enriched in similar extents), the APEX-first procedure is more specific, as it fails to capture the false positive marker GRSF1 (a mitochondrial RBP) whereas the FA-first procedure captures it. This is consistent with our

previous APEX-RIP¹⁶ study and we believe it reflects the fact that FA-first compromises membrane integrity, enabling APEX-generated radicals to escape and non-specifically tag distal proteins.

Figure R9. Comparison of the APEX-first versus FA-first protocol in APEX-PS for enriching nuclear RBPs. The enriched materials were blotted against nuclear RBPs (SRSF1 and hnRNPC), a mitochondrial RBP (GRSF1) and a nuclear non-RBP (ETS2). This data is shown as **Supplementary Figure 2f** in the revised manuscript.

“5. Related to lines 77-78: “non-membrane enclosed regions that are impossible to purify by biochemical fractionation – including the nucleolus and outer mitochondrial membrane (OMM)”, the statement is not entirely accurate since nucleolus has been previously purified biochemically (PMID: 11790298)”

Thank you for pointing this out. We have edited this sentence to: “non-membrane enclosed regions that are difficult or impossible to purify by biochemical fractionation – including the nucleolus and outer mitochondrial membrane (OMM)”

“6. Related to lines 231-232: “We applied the TMT ratio cutoff that maximized retention of true positives and minimized retention of false positives for each dataset’. This statement is not entirely accurate, as in theory, the maximization of TP retention and the minimization of FP retention are two contradictory goals. In a ROC analysis, one typically wants to maximize the difference between TP and FP rates.”

We have edited this sentence to improve its accuracy, per the reviewer’s suggestion.

“7. Related to lines 240-241: “The second approach we used to filter the MS data consists of two-sample T-tests to compare APEX-NLS replicates to background controls” and Supplementary Fig. 5a. Please clarify “background controls”.”

We have clarified that background controls refer to the –APEX and –H₂O₂ controls.

“8. Related to lines 435-438: “To determine if these 11 mRNAs are localized to the OMM solely through the action of SYNJ2BP or if other binding interactions also play a role, we used

OMM-localized APEX to directly biotinylate the RNAs at the OMM in both wild-type and SYNJ2BP knock-out HEK293T cells”, the effect of SYNJ2BP KO can demonstrate that SYNJ2BP participates in the OMM targeting of these mRNAs, but cannot tell whether SYNJ2BP alone or together with other interacting proteins anchors these mRNAs at the OMM. For example, SYNJ2BP may function as a protein anchor for other RBPs. This point could be discussed briefly in the main text.”

Thank you for pointing this out. We cannot rule out the cooperation of other proteins with SYNJ2BP. We have edited the text to clarify this point.

“9. In Supplementary Fig. 5f, “Not annotaed” should be “Not annotated”.

10. Line 431: “SYNJBP” should be “SYNJ2BP”.

11. Lines 313-314: “where proteins are extensively and dynamically interact with RNAs” should be “where proteins are extensively and dynamically interacting with RNAs”.”

Thank you for pointing these out. We have corrected them all.

Reference

1. Fazal, F.M. *et al.* Atlas of Subcellular RNA Localization Revealed by APEX-Seq. *Cell* **178**, 473-490.e426 (2019).
2. Hao, Z. *et al.* N(6)-Deoxyadenosine Methylation in Mammalian Mitochondrial DNA. *Mol. Cell* **78**, 382-395.e388 (2020).
3. Pillon, N.J. & Gabriel, B.M. Transcriptomic profiling of skeletal muscle adaptations to exercise and inactivity. *Nat. Commun.* **11**, 470 (2020).
4. Gagnon, K.T., Li, L., Janowski, B.A. & Corey, D.R. Analysis of nuclear RNA interference in human cells by subcellular fractionation and Argonaute loading. *Nat. Protoc.* **9**, 2045-2060 (2014).
5. Simon, M.D. *et al.* The genomic binding sites of a noncoding RNA. *Proc. Natl Acad. Sci. USA* **108**, 20497-20502 (2011).
6. Chu, C., Qu, K., Zhong, F.L., Artandi, S.E. & Chang, H.Y. Genomic maps of long noncoding RNA occupancy reveal principles of RNA-chromatin interactions. *Mol. Cell* **44**, 667-678 (2011).
7. Chu, C. *et al.* Systematic discovery of Xist RNA binding proteins. *Cell* **161**, 404-416 (2015).
8. Panhale, A., Richter, F.M. & Ramírez, F. CAPRI enables comparison of evolutionarily conserved RNA interacting regions. *Nat. Commun.* **10**, 2682 (2019).
9. Rhee, H.W. *et al.* Proteomic mapping of mitochondria in living cells via spatially restricted enzymatic tagging. *Science* (New York, N.Y.) **339**, 1328-1331 (2013).
10. Hung, V. *et al.* Proteomic mapping of the human mitochondrial intermembrane space in live cells via ratiometric APEX tagging. *Mol. Cell* **55**, 332-341 (2014).
11. Branon, T.C., *et al.* Efficient proximity labeling in living cells and organisms with TurboID. *Nat. Biotechnol.* **36**, 880-887 (2018).
12. Li, Y. *et al.* A Clickable APEX Probe for Proximity-Dependent Proteomic Profiling in

- Yeast. *Cell Chem. Biol.* **27**, 858-865.e858 (2020).
13. Cijssouw, T., et al. Mapping the Proteome of the Synaptic Cleft through Proximity Labeling Reveals New Cleft Proteins. *Proteomes*. **6**, 48 (2018).
 14. Queiroz, R.M.L. & Smith, T. Comprehensive identification of RNA-protein interactions in any organism using orthogonal organic phase separation (OOPS). *Nat. Biotechnol.* **37**, 169-178 (2019).
 15. Keilhauer, E.C., Hein, M.Y. & Mann, M. Accurate protein complex retrieval by affinity enrichment mass spectrometry (AE-MS) rather than affinity purification mass spectrometry (AP-MS). *Molecular & cellular proteomics : MCP* **14**, 120-135 (2015).
 16. Kaewsapsak, P., Shechner, D.M., Mallard, W., Rinn, J.L. & Ting, A.Y. Live-cell mapping of organelle-associated RNAs via proximity biotinylation combined with protein-RNA crosslinking. *eLife* **6** (2017).

REVIEWERS' COMMENTS

Reviewer #1 (Remarks to the Author):

In this revised manuscript the authors have largely addressed my concerns. The authors have made fair arguments justifying data that I had previously questioned and addressed other experimental suggestions in an acceptable manner.

I appreciate the authors' efforts to address my concerns regarding the translation-dependent RBPs and for revising their manuscript to help clarify the selection criteria for hits from their RIP-seq data. Regarding the new data examining OXPHOS components, this data is only suggestive of the conclusion that assembly is affected in the SYNJ2BP KO during recovery following PUR stress. Quantifying complexes under native conditions would be required to fully draw this conclusion. Nonetheless, this new data is informative as it provides a context for comparing CIII and CIV in relation to the other ETC complexes.

Overall, I given these revisions, I find this work is suitable for Nature Communications.

Reviewer #2 (Remarks to the Author):

The authors have satisfactorily addressed my concerns, and I recommend publication of this exciting work.

Reviewer #3 (Remarks to the Author):

The authors have provided detailed responses to all the questions raised. I don't have further questions, but there remains a few minor concerns that the authors should address.

1. There lacks a detailed experimental procedure related to Supplementary Figure 9d. Please clarify the meaning of "fold change of biotinylation".
2. Related to line 414-415, "Interestingly, several proteins (8 out of 29) also have literature connections to mitochondria-ER contact sites", I think it should be 8 out of 28.
3. Related to Figure 7c, "Recvoery" should be Recovery.

Point-by-point responses to reviewer comments

Below are our point-to-point responses to the questions and concerns raised by the reviewers (colored in blue).

Reviewer #1 (Remarks to the Author):

In this revised manuscript the authors have largely addressed my concerns. The authors have made fair arguments justifying data that I had previously questioned and addressed other experimental suggestions in an acceptable manner.

I appreciate the authors' efforts to address my concerns regarding the translation-dependent RBPs and for revising their manuscript to help clarify the selection criteria for hits from their RIP-seq data. Regarding the new data examining OXPPOS components, this data is only suggestive of the conclusion that assembly is affected in the SYNJ2BP KO during recovery following PUR stress. Quantifying complexes under native conditions would be required to fully draw this conclusion. Nonetheless, this new data is informative as it provides a context for comparing CIII and CIV in relation to the other ETC complexes.

Overall, I given these revisions, I find this work is suitable for Nature Communications.

We appreciate the positive evaluation from the reviewer.

Reviewer #2 (Remarks to the Author):

The authors have satisfactorily addressed my concerns, and I recommend publication of this exciting work.

We thank the reviewer for supporting publication.

Reviewer #3 (Remarks to the Author):

The authors have provided detailed responses to all the questions raised. I don't have further questions, but there remains a few minor concerns that the authors should address.

We appreciate the reviewer's comments.

1. There lacks a detailed experimental procedure related to Supplementary Figure 9d. Please clarify the meaning of "fold change of biotinylation".

Thanks for pointing this out. We have added more descriptions in the corresponding figure legend to explain the procedure and the meaning of "fold change of biotinylation". The relative APEX-PS-OMM enrichment (fold change of biotinylation) of +PUR vs -PUR samples was determined by comparing 129C&130N against 127C&128N.

2. Related to line 414-415, "Interestingly, several proteins (8 out of 29) also have literature connections to mitochondria-ER contact sites", I think it should be 8 out of 28.

We have corrected it.

3. Related to Figure 7c, "Recvoery" should be Recovery.

We have corrected it.